

# Particle energy partitioning and transverse diffusion during rarefied travel on an experimental hillslope

Sarah G. W. Williams[1] and David J. Furbish[1]

[1]Department of Earth and Environmental Sciences, Vanderbilt University, Nashville, Tennessee, USA

**Correspondence:** Sarah Williams (sarah.g.williams@vanderbilt.edu)

**Abstract.** Recent theoretical and experimental work (Furbish et al., 2020a, 2020b) indicates that rarefied particle motions on rough hillslope surfaces are controlled by the balance between gravitational heating of particles due to conversion of potential to kinetic energy and frictional cooling of the particles due to collisions with the surface. Here we elaborate how particle energy is partitioned between kinetic, rotational, and frictional forms during downslope travel using measurements of particle travel distances on a laboratory-scale hillslope, supplemented with high-speed imaging of drop-impact-rebound experiments. The drop-impact-rebound experiments indicate that particle shape has a dominant role in energy conversion during impact with a surface. Relative to spherical and natural rounded particles, angular particles give greater variability in rebound behavior resulting in more effective conversion of translational to rotational energy. The effects of particle shape on energy conversion are especially pronounced on a sloping sand-roughened surface. Angular particles travel shorter distances downslope than rounded particles though travel distance data for both groups are well fit by generalized Pareto distributions. Moreover, particle-surface collisions during downslope motion lead to a transverse random-walk behavior and transverse particle dispersion. Transverse spreading increases with surface slope as there is more available energy to be partitioned into the downslope or transverse directions during collision due to increased gravitational heating. Rounded particles exhibit greater transverse dispersion than angular particles, as less energy is lost during collision with the surface. Because the experimental surface is relatively smooth, this random-walk behavior represents a top-down control on the randomization of particle trajectories due to particle shape, which is in contrast to a bottom-up control on randomization of particle trajectories associated with motions over rough surfaces. Importantly, transverse particle diffusion during downslope motion may contribute to a cross-slope particle flux, and likely contributes to topographic smoothing of irregular hillslope surfaces such as scree slopes.

## 1 Introduction

Recent descriptions of sediment transport on hillslopes involving long-distance particle motions have focused on nonlocal transport, where the particle flux at a hillslope position $x$ depends on upslope conditions that influence the entrainment and motions of particles reaching $x$ (Furbish and Haff, 2010; Furbish and Roering, 2013; Doane et al., 2018). Moreover, these descriptions are designed to accommodate rarefied particle motions that do not satisfy the continuum assumption. Namely, during rarefied transport, particle behavior is influenced far more by particle-surface interactions than by particle-particle inter-actions (Furbish et al., 2020a), analogous to granular shear flows at high Knudsen number (Kumaran, 2005, 2006). Describing





nonlocal, rarefied transport is probabilistic in nature, and emerging methods for describing particle motions (Furbish and Haff, 2010; Foufoula-Georgiou et al., 2010; Tucker and Bradley, 2010; Furbish and Roering, 2013) hark back to the pioneering work of Einstein (Einstein, 1938), who conceptualized bed load transport as a probabilistic problem.

To date, probabilistic formulations have mostly involved kinematic descriptions of particle motions and transport with limited elucidation of the associated mechanics. Nonetheless, key elements of these formulations — the particle entrainment rate and the probability distribution of travel distances — provide the basis for connecting the probabilistic formulations with the associated mechanics of particle disentrainment. By explicitly including the distribution of particle travel distances — which depend on forces acting on the particles, particle characteristics such as size and shape, and the surface over which the particles move — nonlocal formulations formally acknowledge the probabilistic nature of sediment motions. Then, by considering the behavior of a great number, or cohort, of particles, it becomes possible to describe the probabilistic physics of sediment transport without necessarily considering the details of individual particle motions (Furbish et al., 2020a).

Nonlocal formulations of transport on hillslopes to this point have focused on downslope travel of particles, neglecting particle motions in two dimensions. Herein we examine transverse (cross-slope) motions in relation to downslope particle motions, extending recent work by Furbish et al. (2020a, 2020b). This work highlights the need for: (1) a conceptualization of how particles interact energetically with the surface over which they move, (2) a demonstration of how particle angularity and size affect downslope and lateral transport distances, and (3) an understanding of the mechanical basis of particle deposition. The experiments presented herein address elements of these problems based on observations of particle motions on a rough inclined surface that is akin to a laboratory-scale hillslope. Building from the work of Furbish et al. (2020a), our focus on particle energetics associated with particle-surface collisions allows us to clarify elements of the mechanics involved in sediment disentrainment and thence the distributions of particle travel distances on hillslopes in relation to the sediment flux.

The purpose of this paper therefore is to describe elements of particle motions on rough hillslopes as a setup for extending current probabilistic descriptions of these motions to two dimensions. We first focus on how particles interact with the surface over which they move, and the effects of these interactions on one-dimensional travel distances. We then turn to transverse components of motion. In Section 2 we describe the essential elements of particle energy extraction during surface collisions, leading to deposition. In Section 3 we summarize the probabilistic theory (Furbish et al., 2020a) concerning rarefied particle motions and disentrainment on rough hillslopes. In Section 4 we describe experiments to clarify the effects of particle angularity and surface roughness on energy extraction during collisions using high-speed imaging of particle drop-impact-rebound experiments. Section 5 describes the second set of experiments involving particle travel distances on the experimental hillslope in relation to particle angularity, size and surface slope. Results and analysis of downslope distances are presented in Section 5.1. In Section 5.2 we describe how transverse particle spreading is connected with downslope travel distances as a result of energy partitioning related to particle angularity and surface slope, and how these motions yield transverse diffusion representing a "top-down" influence of angularity. Thus, our work further unfolds elements of the particle energy balance in relation to particle travel in two dimensions.





## 2   Problem Statement

The initial phase of work presented here and in several companion papers (Furbish et al., 2020a, 2020b) is aimed at clarifying the mechanics of particle disentrainment. To do so, we conducted experiments primarily concerned with rarefied motions. In a sediment transport and geomorphic context, particle transport such as this readily describe rockfall onto scree or talus slopes (Kirkby and Statham (1975), Furbish et al., 2020a) or the dry ravel of particles following disturbance (Gabet, 2003; Roering and Gerber, 2005; Doane, 2018; Roth et al., 2020) or release from sediment capacitors such as vegetation (Lamb et al., 2011,

2013; DiBiase and Lamb, 2013; DiBiase et al., 2017; Doane et al., 2018, 2019). Rarefied transport is fundamentally a stochastic process as each instance of energy extraction depends on the mass, energy state and angularity of the particle. Describing the motions of many particles and their likelihood of deposition is thus a statistical mechanics problem.

By focusing experiments on the rarefied transport of particles down hillslopes of unchanging characteristics (e.g., slope, surface roughness), we are able to observe the stochastic nature of particle-surface collisions and associated two-dimensional

travel demonstrated by this simple system. In this problem, stochasticity is directly due to the physical characteristics of both the particle and the sloping surface. In natural systems, more complex topography with varying slopes and roughness features, including sediment capacitors which disrupt motions (Furbish et al., 2009; Lamb et al., 2013; Doane, 2018) further introduce opportunities for randomizing motion. Here, we neglect these complicating features in order to directly observe and describe the fundamental motions of particles. To develop our understanding of stochastic particle behavior, we first focus on a quantitative

understanding of particle-surface interactions as the energy exchanged and partitioned during these interactions determines the particle travel distances in both the $x$ and $y$ directions.

Consider the motion of a sediment particle dropped onto a rough surface inclined at an angle $\theta$ at position $x = 0$. The motion of the particle is similar to those of previously released particles (Figure 1). The potential energy of the particle, with a maximum value at the initial position $x = 0$, is converted to kinetic energy (hereafter referred to as gravitational heating)

as it moves downslope. Frictional cooling acts to counter this heating as the particle bounces down the rough surface. This cooling is due to an extraction of translational energy during short duration collisions with the surface. Energy is also lost to minute deformation of the particle and surface during collisions. The clickety-clack sounds emitted by the particles as they move downslope is a sonic manifestation of the irreversible energy loss taking place (Furbish et al., 2020b). If the rate of heating outweighs the rate of cooling, the particle continues to move down the slope. The particle is disentrained, or deposited,

at the point on the surface where heating is overcome by cooling through collisional friction. In our experiments, described below, we do not observe sliding, suggesting that Coulomb-like frictional cooling is a negligible part of the energy balance of particles being transported down a hillslope. Therefore, the problem consists of verifying the partitioning of the translational kinetic energy into direct losses and other forms of energy during collision, including deformation, rotation, and transverse translation. The distances that particles travel following entrainment directly reflect the probabilistic mechanics of motion and

deposition.



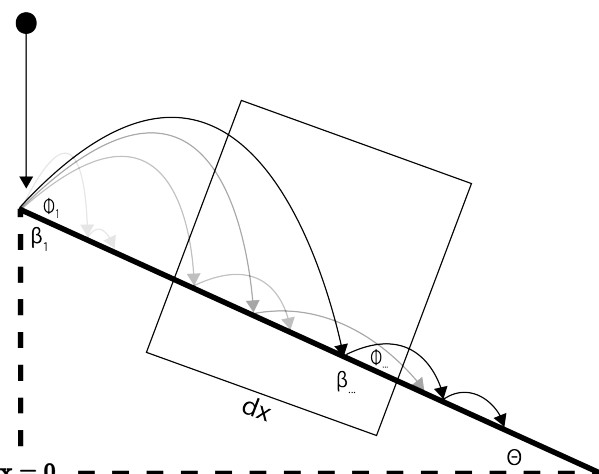

**Figure 1.** Diagram depicting particle-bed collisional energy extraction ($\beta$) and angle of reflection ($\phi$) on an inclined plane of specified angle $\theta$. Particle path in black is one example of many possible paths. Grey lines depict paths of similar particles. Note the lack of sliding or rolling in this schematic.

## 3 Relevant Theory

Let $x$ denote the downslope coordinate parallel to the surface (Figure 1) and let $y$ denote the horizontal transverse coordinate. The rarefied transport of sediment on hillslopes involves motions in both coordinate directions such that the particle flux and associated changes in the land-surface elevation at a position $(x, y)$ may involve transverse motions of particles starting at positions different from $y$. If $r$ and $s$ denote particle travel distances parallel to $x$ and $y$, respectively, then $f_{r,s}(r, s; x, y)$ denotes the joint probability density function for these displacements $r$ and $s$. We consider elements of this two-dimensional transport in several sections below, and provide a general treatment of the entrainment forms of the two-dimensional flux and the Exner equation in Appendix A as context. Meanwhile, for simplicity we focus on the one-dimensional versions of the entrainment form of the Exner equation and the volumetric particle flux to clarify key ingredients of rarefied particle transport on hillslopes. In this situation the probabilistic formulations of sediment transport start with the one-dimensional entrainment form of the Exner equation (Tsujimoto, 1978; Parker et al., 2000) ,

$$c_b \frac{\partial \eta(x,t)}{\partial t} = -E(x,t)$$

$$+ \int\limits_{-\infty}^{x} E(x',t) f_r(x - x'; x', t)\, \mathrm{d}x', \tag{1}$$





where $E(x,t)$ is the volumetric entrainment rate at position $x$ and time $t$, $f_r(r;x,t)$ is the distribution of the travel distances
$r$ of particles whose motions start at position some upslope position $x'$. The cumulative distribution function is denoted as
$F_r(r;x,t)$. In turn, the volumetric flux parallel to $x$ is

$$q(x,t) = \int_{-\infty}^{x} E(x',t) R_r(x - x'; x', t)\, \mathrm{d}x',\qquad(2)$$

where $R_r(r;x,t) = 1 - F_r(r;x,t)$ is the associated exceedance probability function. Moreover, because Eq. (1) and Eq. (2) are

nonlocal and scale independent, these expressions are applicable to particle motions and associated mechanics in a variety of
settings (Foufoula-Georgiou et al., 2010; Furbish and Haff, 2010; Furbish and Roering, 2013).

    Returning to the simple hillslope (Figure 1), consider the motions of many particles entrained at position $x = 0$. The particle
travel distance $r$ as defined above now may be recast a $x$ such that the probability density function $f_r(r;x)$ becomes $f_x(x)$.
Imagine a control volume of dimension $\mathrm{d}x$ parallel to the mean downslope particle motions. Over some period of time a great

number of particles is released from position $x = 0$ and many enter the left face of the control volume. Some of the particles that
enter the control volume exit through the right face after traveling the interval $\mathrm{d}x$. Some of the initial great number of particles
released do not reach the control volume before coming to rest while other particles come to rest within the volume. Many, if
not all, of the particles that reach the left face of the control volume interacted with the surface before exiting the right face.
If the great number of particles are treated as a cohort, independent of time (Appendix B of Furbish et al., 2020a), then $N(x)$

denotes the number of particles that enter the control volume and $N(x+\mathrm{d}x)$ denotes the number of particles that exited the
volume. Thus, the number of particles deposited, or disentrained, in the small interval $x$ to $x + \mathrm{d}x$ is $\mathrm{d}N = N(x+\mathrm{d}x) - N(x)$.
The spatial rate of particle deposition in the interval is $\mathrm{d}N/\mathrm{d}x$ and depends on the energy states of the particle in the cohort.

    Each particle in the cohort has mass $m$ and downslope velocity $u$ such that $E_p = (m/2)u^2$ denotes its translational kinetic
energy. The arithmetic average of the energy for the great number of particles $N$ is $E_a = \langle E_p \rangle$ where angle brackets denote

an ensemble average of particle energy states. The total energy of the system, which must satisfy laws of conservation, is then
$E = N E_a$. With the full formulation developed in Furbish et al. (2020a), the conservation of the total energy of the cohort is
given by

$$\frac{\mathrm{d}E(x)}{\mathrm{d}x} = Nmg\sin\theta - Nmg\mu\cos\theta - \frac{1}{\alpha}Nmg\mu\cos\theta.\qquad(3)$$

The terms on the right side of Eq. (3) represent gravitational heating, frictional cooling due to particle-surface collisions, and

energy loss associated with deposition, respectively. The ratio of gravitational heating to frictional cooling is defined by the
dimensionless Kirkby number $Ki$, which may be written as

$$Ki = \frac{4\tan\phi S}{\langle \beta_x \rangle},\qquad(4)$$

where $\phi$ denotes the expected reflection angle of particles following collision with the surface, $S$ denotes the magnitude of the
surface slope, and $\beta_x$ denotes the proportion of the translational particle energy $E_p$ extracted by the collision, namely,

$$\beta_x = -\frac{\Delta E_p}{E_p}.\qquad(5)$$





If we write the Kirkby number as $Ki = S/\mu$, then

$$\mu = \frac{\langle \beta_x \rangle}{4 \tan \phi} \tag{6}$$

may be considered a friction coefficient. We emphasize that the frictional cooling that a particle experiences as it travels downslope is not to be interpreted as a result of Coulomb-like dynamic friction (Kirkby and Statham, 1975; Gabet and Mendoza, 2012; DiBiase et al., 2017) but rather is a result of particle collision with the surface. In turn, for a given initial average particle energy $E_{a0}$, a characteristic length scale of deposition $L_c$ can be defined as

$$L_c = \alpha \frac{E_{a0}}{mg\mu\cos\theta}, \tag{7}$$

where the factor $\alpha$ modulates this length scale, likely in relation to particle size, angularity and mode of motion (e.g., translational versus rotational). Here it becomes clear that collisional friction, via $\mu$ in the denominator of Eq. (7), contributes to setting the expected distances of particle motions.

Focusing specifically on the rarefied particle motions described above and visualized in Figure 1, theory developed in Furbish et al. (2020a) defines the disentrainment rate function as

$$P_x(x) = \frac{1}{Ax + B} \tag{8}$$

where $A \in \Re$ is a shape parameter and $B > 0$ is a scale parameter. These distribution parameters may be represented in mechanistic terms as

$$A = \frac{\alpha}{\gamma} \left[ \frac{S}{\mu} - 1 + \frac{1}{\alpha}(\gamma - 1) \right] \qquad \text{and} \tag{9}$$

$$B = \frac{\alpha}{\gamma} \frac{E_{a0}}{mg\mu cos\theta} \tag{10}$$

where $\gamma$ is the ratio of the arithmetic average energy to the harmonic average energy (Furbish et al., 2020). Based on Eq. (8), the downslope travel distances have a probability density function $f_x(x)$ with the form of a generalized Pareto distribution (Figure 2). This probability density function of downslope travel distances with a position parameter equal to zero may be written as

$$f_x(x) = \frac{B^{1/A}}{(Ax + B)^{1+1/A}}. \tag{11}$$

The cumulative distribution function of travel distances is

$$F_x(x) = \begin{cases} 1 - \frac{B^{1/A}}{(Ax+B)^{1+1/A}} & A \neq 0 \\ 1 - e^{-x/B} & A = 0 \end{cases} \tag{12}$$

and the associated exceedance probability is

$$R_x(x) = \begin{cases} \frac{B^{1/A}}{(Ax+B)^{1/A}} & A \neq 0 \\ e^{-x/B} & A = 0 . \end{cases} \tag{13}$$





Note in the formulation above that surface conditions, that is, slope angle and surface roughness, are assumed to be uniform across the system. This may be modified for nonuniform downslope conditions (Furbish et al., 2020a).

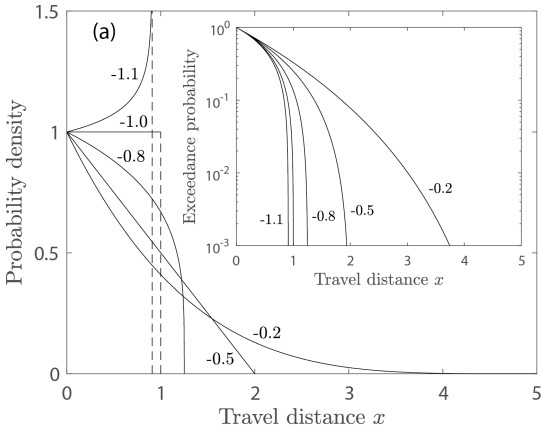
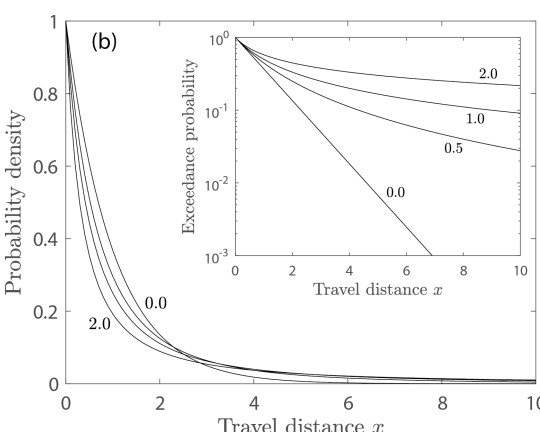

**Figure 2.** Plot of the generalized Pareto distribution of particle travel distances $x$ for scale parameter $B = 1$ and different values of the shape parameter $A$ with (a) $A < 0$ representing rapid thermal collapse and (b) $A \geq 0$ with associated exceedance probability plot (inset) representing net heating of particles. Compare with Figure 1 in Hosking and Wallis (1987).

The specific form of this distribution is determined by the magnitude of the Kirkby number. A small value of $Ki$ leads
to rapid thermal collapse represented by a bounded distribution of travel distances. An intermediate value of $Ki$ leads to isothermal conditions with an exponential distribution of distances. Large $Ki$ leads to net heating of particles represented by a heavy-tailed form of the Pareto distribution. In particular, data suggest that large particles on average travel farther than small particles for given surface slope and roughness conditions (DiBiase et al., 2017), and experiments indicate that rounded or spherical particles stay in motion longer than angular particles. What remains unclear is how particle size and angularity
influence the expected proportion $\langle \beta_x \rangle$ of energy extracted during collisions, and how these particle qualities might influence the factors $\alpha$ and $\mu$, specifically by increasing or decreasing the likelihood of deposition, including possible effects of rotational motion. The experiments described in the following sections were designed to clarify these elements of particle motions and deposition.

## 4 Particle energy extraction with collision

The quantity $\beta_x$ in Eq. (5) is nominally related to a coefficient of restitution $\epsilon_x$ as $\beta_x = 1 - \epsilon_x^2$. However, the change in translational energy $\Delta E_p$ is partitioned between deformational friction, rotational energy and transverse motion, so the coefficient $\epsilon_x$ (and therefore the factor $\beta_x$) cannot simply represent a coefficient of restitution – although particle collision theory suggests that this coefficient includes effects of normal and tangential coefficients of restitution as normally defined (Brach, 1991;





Stronge, 2000). This means that $\beta_x$ must be treated formally as a random variable rather than a fixed deterministic quantity as
in granular gas theory. The experiments described next are designed to clarify the elements and behavior of $\beta_x$.

## 4.1  Experiments

The first set of experiments are aimed at demonstrating the basis for treating the proportion of energy extraction, $\beta_x$, as a
random variable using simple particle drop experiments. To do this, experiments focused on the analogous quantity $\beta_z$, which
is the amount of energy extracted in the $z$-direction following vertical free fall onto a horizontal surface. Focusing on this
quantity allows us to observe and calculate energy partitioning with collision. Gravel sized particles (Figure 3) were dropped
onto both a smooth slate surface and a rigid concrete surface with sand-scale roughness. Particle angularity varied, with two
groups composed of rounded and angular particles based on visual inspection. Drop heights above each respective surface
included 8 cm, 12 cm, 16 cm, and 20 cm. In addition, spherical glass marbles were dropped during several of the experiments
for comparison with the natural particles. Each particle motion was recorded with a Lightning RDT monochrome camera (DRS
Technologies) operating at 800 frames per second with an image resolution of 1,280 × 640 pixels. Particle rebound energy
was determined using the amount of time between the first and second collision related to the maximum height of a parabolic
rebound trajectory (see below). The data reported here involve significantly more observations than described in Furbish et al.
(20202020).

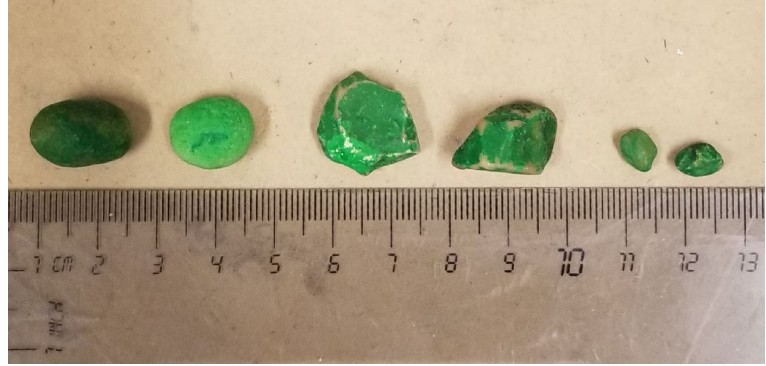

**Figure 3.** Representative samples of natural rounded particles (left), natural angular particles (center), and natural small particles of mixed
angularity (right) used in experiments. Particles were hand selected and care was taken to avoid picking only the "prettiest", most well shaped
particles. Glass marbles, of similar size to the rounded and angular particle groups, are not shown.

## 4.2  Results

Cumulative probability plots of $\beta_z$ for the marbles, rounded natural particles and angular natural particles (Figure 4) reveal
that the shape of the particles dramatically affects energy loss with collision and resulting rebound for all drop heights. Slight
variations in the cumulative plots with changing height (not shown) were likely attributable to randomness associated with the
small number of experiments conducted for each height, so data presented here are pooled simply into shape groups. Each of





the six particles was dropped 12 times resulting in 72 data points for each height and surface combination. When results are
pooled, each surface has more than 300 observations.

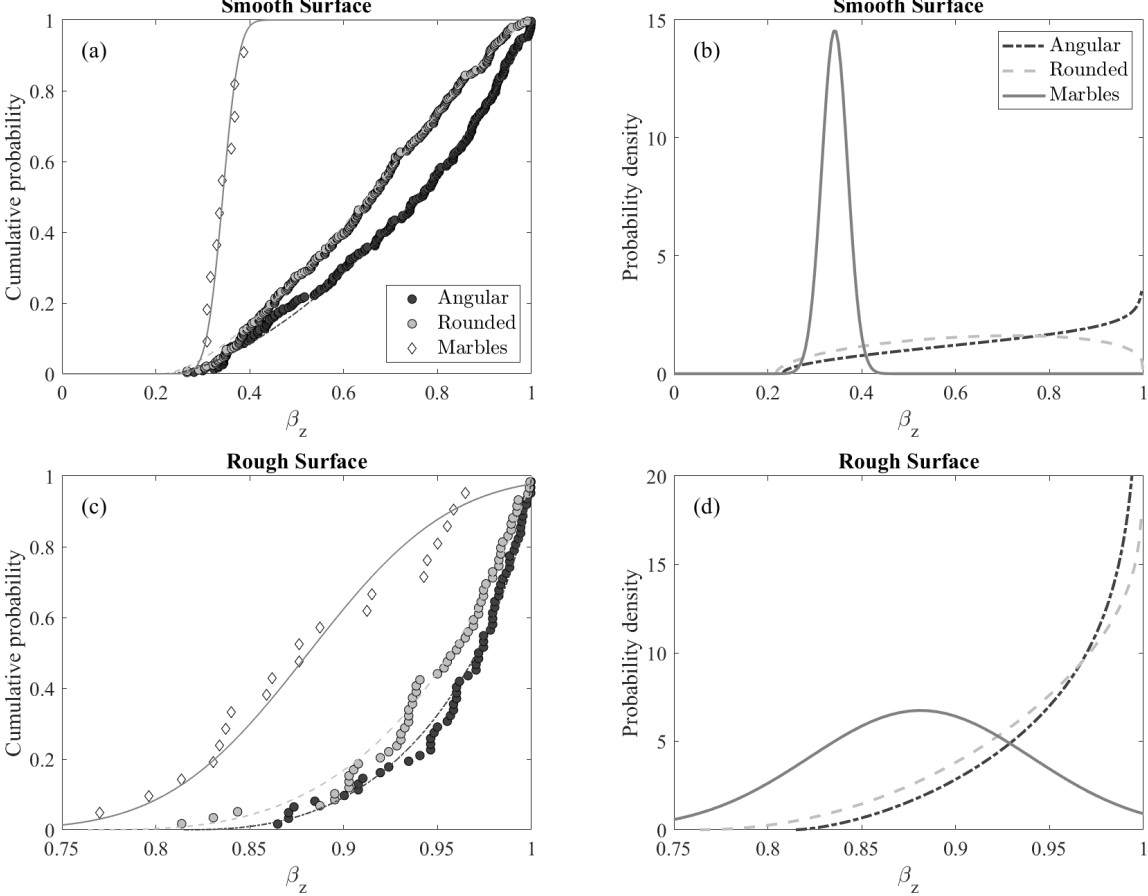

**Figure 4.** Drop experiment results of $\beta_z$ for glass spheres fit to a Gaussian distribution and rounded and angular gravel particles fit to a beta distribution shown with (a) cumulative distribution plot for a smooth surface, (b) probability density plot for a smooth surface, (c) cumulative distribution plot for a rough surface, and (d) probability density plot for a rough surface.

The data for spheres are fit with a Gaussian distribution while the data for natural particles are fit with beta distributions. Independent evidence indicates that the distribution of $\epsilon$ for sphere takes the shape of a Laplacian probability density function (Gunkelmann et al., 2014). We stress that the beta distribution was selected based on parameter properties that well fit the data, not on mechanistic underpinnings. The small variance in $\beta_z$ for the marbles is likely due to the highly collinear collisions of the
marble with the smooth surface with only small variations in the coefficient of restitution. These variations likely result from small imperfections on the surfaces and deformations resulting from collision. In comparison, the probability distributions of energy extraction with collision for both rounded and angular natural particles have larger variance on the smooth surface (Figure 4b) than on the rough surface (Figure 4d). Note the changes in axes in these figures. Spherical marbles have a maximum



$\beta_z$ value of about 0.35 for the smooth surface compared to a maximum $\beta_z$ value of 0.87 for the rough surface. For all particle
shapes, the associated probability density functions and cumulative probability distributions (and mean values) are strongly
shifted towards unity (Figures 4c,4d). Probability distributions for natural rounded and natural angular particles plot closely to
one another and show much less variance in energy extracted with collision on the roughened surface compared to spherical
marbles (Figure 4d).

The percentage of recovered energy for each angularity and surface arrangement are recorded in Table 1, where the coef-
ficient of restitution $\epsilon$ has been calculated based on the longest rebound time. For each particle dropped onto the horizontal
surfaces, the initial impact energy is denoted as $E_{p0}$. During collision, this initial energy is partitioned into vertical and hori-
zontal translational energy, frictional loss $f_c$ in the form of heat, sound, or deformation, and rotational kinetic energy. For each
set of particle shape and surface roughnesses in Table 1, frictional loss $f_c$ is approximated to equal $(1-\epsilon^2)E_{p0}$. The converted
translational energy associated with surface parallel motion and rotational energy following collision, $E_c$, is thus determined
to be

$$E_c = \epsilon^2 E_{p0} - \frac{1}{8} m g^2 T^2 \tag{14}$$

where $T$ is the travel time between collisions (Furbish et al., 2020b). Here, $\epsilon$ is intended to be the true coefficient of restitution.
We offer estimations of this coefficient based on our data with longest observed rebounds used in the calculations, as these
rebounds are closest to being collinear-like with the least amount of energy lost during collision.

| Category | Surface | $N$ | $\epsilon$ | $f_c$ | $E_c$ | Recovered |
|---|---|---|---|---|---|---|
| Angular | Slate | 234 | 0.82 | 0.33 | 0.37 | 28 % |
| | Concrete | 61 | 0.37 | 0.86 | 0.10 | 4 % |
| Rounded | Slate | 280 | 0.82 | 0.32 | 0.33 | 35 % |
| | Concrete | 59 | 0.43 | 0.81 | 0.13 | 5 % |
| Marbles | Slate | 20 | 0.81 | 0.34 | 0.03 | 66 % |
| | Concrete | 20 | 0.33 | 0.89 | 0.02 | 11% |

**Table 1.** Average particle energy partitioning following first collision with a surface as a proportion of initial energy $E_{p0} = mgh$. The
normal coefficient of restitution $\epsilon$ was determined using the highest rebound height for each angularity-surface pairing. Calculations of
average frictional loss $f_c$ and converted translational energy $E_c$ are based on pooled data for each particle shape and surface roughness
across all recorded heights.

The shift in $\beta_z$ with particle category is interpreted to be a result of the likelihood of collinear collisions, which depends on
the angularity of a particle as well as the roughness of the surface on which the impact occurs. When a collision is collinear the
center of mass is directly situated over the point of contact, which has a negligibly small size, resulting in no torque about the





center (Lim and Stronge, 1994). As one may expect, the likelihood of a collinear collision decreases the more oddly-shaped a particle becomes. To visualize this, consider a sphere with no initial rotation dropped onto a smooth surface (Figure 5). This

system has infinite collinear geometries as every point on the sphere surface can collide in a manner such that the center of mass is in line with the contact point during collision. Under frictionless conditions, rotation of the sphere would be negligible and the rebound would remain in line with the center of mass and contact point (Lim and Stronge, 1994). Now imagine the sphere has some intial rotation during free fall. If either the rotating sphere or the surface are roughened, the center of mass remains in line with the contact point but the rebound will not be collinear as the spin of the sphere produces a finite frictional

torque resulting in a change of rebound path. A highly roughened surface precludes the occurrence of collinear geometries, as contact with the surface may involve multiple points and micro-asperities are unlikely to give in-line impacts. Collisions will thus result in some amount of rotational energy loss.

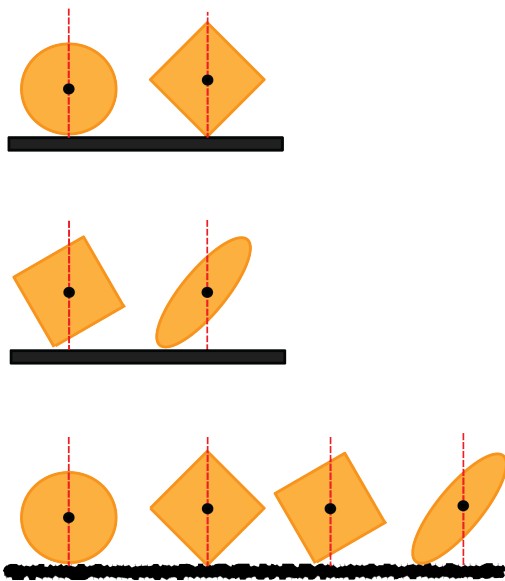

**Figure 5.** Illustration of collinear and non-collinear collisions for particles with center of mass depicted in black and line of collision depicted in red. Particle shape and orientation affects likelihood of collinear collision as does surface roughness.

Consider a cube for comparison. Assume the surface of the cube and the surface on which it is dropped are frictionless. Geometrically the cube has 26 possible perfect collinear geometries (8 corners, 6 faces, 12 edges). The 26 possible geometries

can be reduced to 8, however, as the likelihood of all points on a plane or a edge colliding on a surface at the precisely the same instant is effectively zero. Only the points of the cube are locations of non-trivial probability for collinear collision. However, imagine that the cube is stretched to form a cuboid. There is still a possibility for collinear collision on one of the corners but the overall shape and mass of the cuboid adds variability to possible motions following collision. For a collision to be collinear all torques about the center must be balanced. The likelihood of a precise collinear collision for a cuboid with uneven

measurements about the center of mass is effectively zero. Surface roughness further decreases the likelihood of collinear





collision and influences the torque on the particle leading to increased conversion of translational energy to rotational energy during surface impact.

Particle shape together with characteristics of the surface and incident conditions of motion determine the rebound motion. For a sphere with acute incident angle to the surface, the collision is non-collinear, but the collision geometry is precisely deter-
mined by the incident angle leading to only one outcome. In contrast, for a non-spherical (angular or rounded) particle with the same incident angle, a great number of outcomes are possible depending on the particle orientation, each outcome involving a different extraction of incident kinetic energy. Several intriguing rebound paths were observed during experimentation although no perfect collinear collisions were observed, as to be expected. Files "Rounded_colinear.avi" and "Angular_colinear.avi" provided in the supplementary materials show examples of near collinear collisions on the hard slate with minimal roughness.
Particle rebounds captured in these videos are essentially without rotation following collision and nearly reach maximum vertical displacement. These two examples, which were used to estimate the coefficient of restitution for the particle-slate collisions seen in Table 1, are in stark contrast to files "Angular_all_rotational.avi" and "Semiangular_rotational_die.avi". Particles in these videos experience a strong conversion of translational energy to rotational energy with collision leading to small rebound heights and relatively short periods of motion between successive collisions. These examples are more characteristic of the
overall behavior of particles during these experiments, especially on the roughened surface. This rapid conversion of translational energy to other forms is evident in the cumulative distribution plots of energy extracted during collision $\beta_z$. Marbles on the smooth surface experience the least energy extracted of the three shape groups, and angular particles experience slightly more energy partitioning than rounded particles. On the roughened surface with effectively zero chance of collinear collision, energy is extracted at a much higher rate for all dropped particles.

## 265 5 Particle travel distances

### 5.1 Experiments

With this view of how particle shape likely influences the partitioning of kinetic energy into rotational energy during particle-surface collisions, we now turn to particle travel distances. As evident from Eq. (4), the ratio of gravitational heating to frictional cooling for a particle moving downslope directly depends on the ensemble average of the random variable $\beta_x$. The goal of
the second set of experiments involves measuring travel distances in relation to particle size and angularity for a uniformly rough surface. These travel-distance trials involved an experimental hillslope consisting of a concrete surface with sand-scale roughness and adjustable incline angle (Figure 6). Particles were launched down the slope with a pendulum catapult device which allowed for a consistent surface-parallel initial velocity and negligible rotational motion. Three different sets of particles were used in the experiments: angular particles with approximate diameters of 1 cm, rounded particles with diameters of 1 cm,
and small particles with approximate diameters of 0.5 cm and mixed angularity (Figure 3). Slopes $S$ used were 0.0, 0.09, 0.15, 0.18, 0.25, and 0.28. Particle travel distances in both the downslope and cross-slope directions were recorded based on final resting positions, and supplementary video of a sampling of initial trajectories and impacts were made using the same camera as the drop experiments with a resolution of $640 \times 640$.





Earth **Surface** Dynamics Discussions

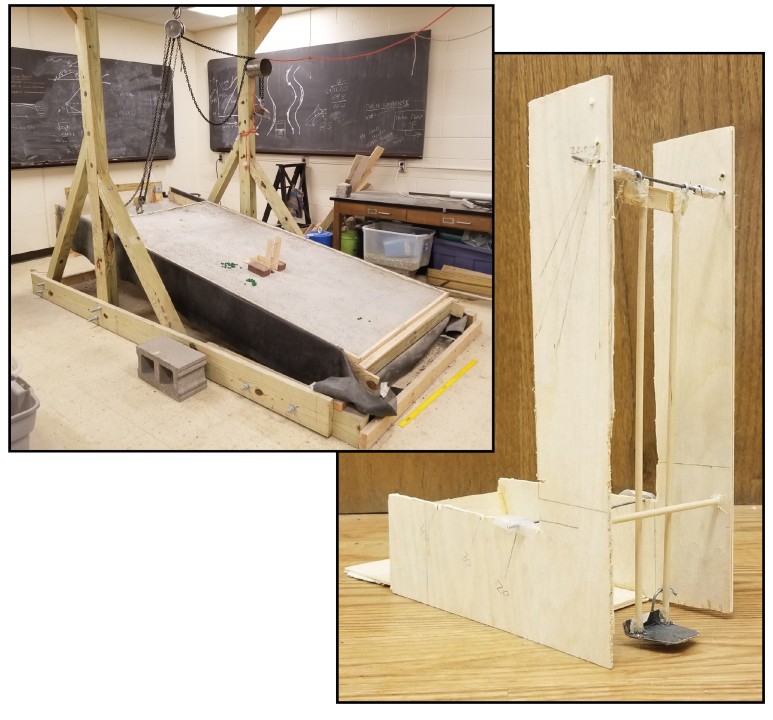

**Figure 6.** Set up for travel distance experiments. Left: The experimental hillslope surface is concrete with sand scale roughness. The angle of the slope can be adjusted with a pulley system. Right: Pendulum catapult constructed to deliver particles to the hillslope surface with negligible rotational motion, minimum velocity perpendicular to slope, and known slope-parallel velocity. A gravel-sized particle is placed on the low-friction (glossy cardboard) cradle surface at the base of the 20 cm pendulum arms. A wand is used to gently push the cradle back to a preset angle then quickly removed. The pendulum would be stopped by the wooden bumper rod installed on the frame thus leaving it up to momentum to deliver the particle to the surface. The cradle rests about 2 mm above the surface at its lowest point.

## 5.2  Downslope travel distance

High-speed imaging of particles launched from the catapult show that particles travel a small distance after launch before their first collision with the surface and experience negligible rotational motion during this flight. The length of this initial flight consistently increases with slope, and particle motion only starts to become randomized by surface collisions following this initial flight, often with the onset of rotational motion. The inflection in the initial exceedance probability plots (not shown) reflect the uniformity of the launch velocities followed by a finite distance over which randomization of the motions

occurs. The inflection does not necessarily directly correspond with the flight distance, and the details of the physics prior to randomization are unclear (Furbish et al., 2020b). For these reasons we truncate the plots at the inflection position, then recalculate exceedance probabilities with reduced $N$ (Appendix B in Furbish et al., 2020b) in order to focus analysis on the randomized particle motions.





**Figure 7.** Plots of exceedance probabilties versus down slope travel distance for experiments over six values of slope $S$. Natural angular (dark grey), natural small (medium grey),and natural rounded (light grey) particles are shown together fitted with generalized Pareto distributions of varying $A$ and $B$ parameters. The concavity of these semi-log plots indicated whether particles are being heated (positive concavity) or cooled (negative concavity). All particles experience rapid thermal collapse on low slopes 0.00 and 0.09 then transition to heavy tailed distributions indicating net heating for slopes 0.25 and greater. Rounded particles travel further downslope before becoming disentrained compared to angular and small particles due to the decreased relative effectiveness of collisional friction. On slopes greater than 0.18, rounded particles do not come to rest on the roughened experimental hillslope surface.





The modified experimental travel distance data are fit with the generalized Pareto distribution and plotted using exceedance
probability plots (Figure 7). Values for the shape and scale parameters $A$ and $B$ (Table 2) were obtained by fitting the data
visually (Furbish et al., 2020a). In addition we used quartile-quartile plots (not shown) to assess consistency with the generalize
Pareto distribution fits. Due to the small sample size, over-fitting of the tails in semi-log plots was avoided (see Appendix A
in Furbish et al. (2020b)). Estimated initial slope-parallel particle velocities $u_0$ were determined using high speed imaging of
particles launches from the catapult (Table 3).

| Category | Slope | $A$ | $B$ (m) | $Ki$ | $\mu$ |
|---|---|---|---|---|---|
| Angular | 0.00 | -0.54 | 0.033 | 0.00 | 0.45 |
|  | 0.09 | -0.39 | 0.075 | 0.23 | 0.40 |
|  | 0.15 | -0.36 | 0.120 | 0.40 | 0.37 |
|  | 0.18 | -0.56 | 0.220 | 0.54 | 0.33 |
|  | 0.25 | 0.30 | 0.350 | 0.98 | 0.25 |
|  | 0.28 | 0.77 | 1.180 | 1.07 | 0.26 |
| Rounded | 0.00 | -0.51 | 0.05 | 0.00 | 0.32 |
|  | 0.09 | -0.24 | 0.12 | 0.36 | 0.25 |
|  | 0.15 | 0.10 | 0.19 | 0.76 | 0.20 |
|  | 0.18 | 0.02 | 0.28 | 0.80 | 0.22 |
| Small | 0.00 | -0.49 | 0.03 | 0.00 | 0.36 |
|  | 0.09 | -0.51 | 0.10 | 0.26 | 0.35 |
|  | 0.15 | -0.35 | 0.12 | 0.39 | 0.38 |
|  | 0.18 | -0.51 | 0.24 | 0.54 | 0.33 |
|  | 0.25 | 0.30 | 0.97 | 0.98 | 0.25 |

**Table 2.** Fitted and estimated parameter values for travel distance experiment data shown in Figure 7. $A$ and $B$ are estimated shape and
scale parameters from data fit by eye with generalized Pareto distributions. The Kirkby number $Ki$ is equal to $S/\mu$ where $\mu$ is the friction
coefficient.

The fits and parameter values presented in Figure 7 and Table 2 are specific cases of the generalized pareto distribution. To
illustrate this idea, we calculate the modified exceedance probability $R_*$ and the dimensionaless travel distance $x_*$ given by
(Furbish et al., 2020b)

$$R_* = R_x{}^A \text{ and } x_* = \frac{A}{B}x + 1.$$ (15)





If consistent with a generalized Pareto distribution, these values should collapse to a straight line with a slope of $-1$ in log-log
space (Figure 8). The deviations of the distribution tails are accentuated using this method of plotting and the censored data
associated with slopes over $0.18$, though used during calculation, are not included in the plot.

| Category | Slope | $u_0\ (m\,s^{-1})$ | $N$ |
|----------|-------|--------------------|-----|
| Angular | 0.00 | $0.58 \pm 0.036$ | 10 |
|         | 0.09 | $0.79 \pm 0.053$ | 5 |
|         | 0.15 | $0.86 \pm 0.041$ | 5 |
|         | 0.18 | $0.84 \pm 0.002$ | 5 |
|         | 0.25 | $1.00 \pm 0.031$ | 5 |
|         | 0.28 | $0.98 \pm 0.036$ | 5 |
| Rounded | 0.00 | $0.60 \pm 0.060$ | 10 |
|         | 0.09 | $0.80 \pm 0.021$ | 5 |
|         | 0.15 | $0.86 \pm 0.050$ | 5 |
|         | 0.18 | $0.88 \pm 0.004$ | 5 |
| Small | 0.00 | $0.55 \pm 0.150$ | 10 |
|       | 0.09 | $0.77 \pm 0.038$ | 5 |
|       | 0.15 | $0.90 \pm 0.038$ | 5 |
|       | 0.18 | $0.91 \pm 0.019$ | 5 |
|       | 0.25 | $0.97 \pm 0.013$ | 5 |

**Table 3.** Slope-parallel velocities leaving the launcher cradle for rounded, angular, and small particles. Velocities were calculated from high speed videos of particles launched at varied slopes.

The angularity of particles launched on the roughened experimental hillslope directly affects the downslope travel distances. High speed imaging of initial particle impacts with the surface show a variety of impact geometries which appear to influence the motions during the subsequent travel of the particles (Supplementary Materials, Vanderbilt University Institu-
tional Repository,http://hdl.handle.net/1803/9742). Natural rounded particles travel further downslope than natural angular and small mixed-angularity particles (Figure 7) at the same low slope angle and similar initial slope-parallel velocities. On slopes $S = 0.00$ to $0.15$, concave-down plots show that particles experience rapid thermal collapse ($A < 0$). Initial energy, and energy gained from modest heating, is rapidly dissipated by collisional friction during randomization of motions, in some cases involving Coulomb-like frictional loss during brief impulses (see the file "Rounded_0slope.avi" in supplementary materials).





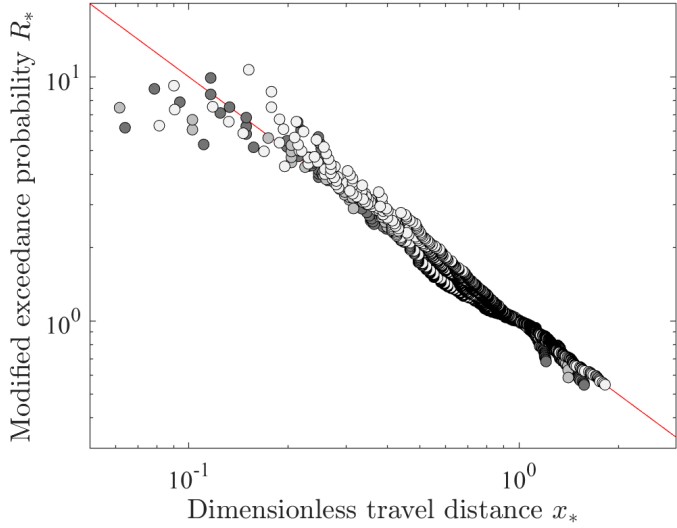

**Figure 8.** Plot of modified exceedance probability $R_*$ versus nondimensional travel distance $x_*$ for angular (dark gray), small(medium gray), and rounded (light gray) particles, and slopes S = 0.00 to S = 0.28. The collapse of these values to a straight line with a slope $-1$ in log-log space indicates consistency with a generalized Pareto distribution.

Following initial contact with the surface, downslope motions appear to transition to bouncing and rotational motions, as shown in files "Angular_18%slope.avi" and "Angular_28%slope.avi". At higher slopes of 0.18, 0.25, and 0.28, the rounded particle distributions transition to a heavy-tailed behavior and eventually particles do not stop on the concrete surface due to net heating ($A > 0$). Downslope travel distances become longer for both angular and rounded particles due to decreasing effectiveness of frictional cooling ($Ki > 1$) with increasing slope, in part due to less frequent particle collision with the surface

and increased conversion of translational energy to rotational energy (Figure 7). All rounded particles exited the slope surface at slopes 0.25 and 0.28. For the same slopes, angular particles are disentrained on the surface, although a significant number of particles reached the wooden bumper at the base of the concrete slope. These censored motions are included in the calculations of the exceedance probabilities, and contribute to the heavy-tailed nature of the generalized Pareto distributions at these slopes.

    The small particles, which were not separated by angularity, suggest that size does not directly affect downslope travel

distance. These small particles are fit with a single distribution, although a mixed distribution may be more appropriate. The small particles experience a behavior in between those of the larger angular and rounded particles, which may be more aptly described by two or more distributions. This possibility suggests that angularity influences travel distance more than size or mass. This is consistent with the idea that mass does not appear in formulation of $Ki$ nor appears in Eq. (5) for $\beta_x$.

### 5.3    Lateral spreading

Particle dispersion is a key element of sediment transport (Samson et al., 1998; Furbish and Haff, 2010; Tucker and Bradley, 2010; Furbish et al., 2012a). Particles starting at the same location, when subjected to the same macroscopic forcing, spread





spatially over time in both laboratory and field settings (Schumm, 1967; Samson et al., 1998). In a simplified way, the spreading behavior of particles on a hillslope is akin to particle behavior on a Galton board, which serves to illustrate the dispersion process due to particle collisions with roughness elements (pegs) attached to the board (Figure 9) (Galton, 1894).

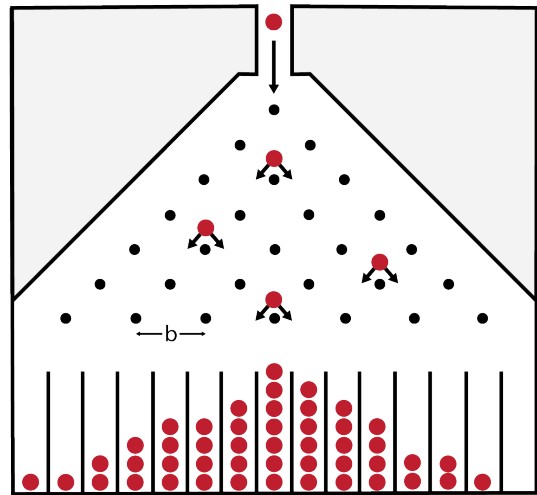

**Figure 9.** Straight-on view of a Galton board with fixed pegs through which particles move illustrating random walks that result a binomial distribution.

Lateral spreading on a Galton board arises as particles roll down a smooth surface under the force of gravity and interact with pegs normal to the surface, which act as roughness elements. An individual peg introduces two degrees of freedom, either right or left, to the sphere's consistent downslope motion. This quasi-random walk of the particles on a Galton board results in a binomial distribution of net transverse displacements, which transitions to a Gaussian distribution with increasing number of peg rows. Mechanical dispersion operates only in the presence of advection due to compelling forces such as fluid drag or, as in

the case of the Galton board, the effect of gravity. In groundwater flow, lateral spreading — mechanical dispersion — is largely dictated by the geometry of the porous medium rather than being attributable to molecular diffusion. When considering the motions of particles on a hillslope, gravitational heating causes particles to be advected downslope but frictional cooling acts to moderate this forcing and randomize the motions. In the case of the Galton board, the motion of the particle is influenced in a bottom-up manner by the roughness elements on the surface.

This surface roughness is only a part of the problem of particle motions on hillslopes, where particle characteristics — in particular angularity — provide a top-down influence on randomizing motion. In our experiments, particles are not interacting with set roughness elements like equally spaced pegs, but rather are experiencing stochastic motions from the outset due to collisions with the randomly-roughened surface in concert with the influence of particle angularity. Natural angularity plays into this randomization of motion, as evident from the results of Section 4.2, as the particles do not mimic the spheres normally

used in Galton boards. The decreasing likelihood of collinear collisions with increased angularity and roughness increases the





degrees of freedom available to the particles. The particle is free to move in any direction although still influenced by gravity. This variation in forces applied to a particle and additional degrees of freedom thus lead to spreading behavior that is more complex than that on a Galton board.

Transverse $y$ positions were measured in addition to downslope $x$ travel distances for angular and rounded particles during
the previous experiments. The simple geometry of the experimental hillslope prevented upslope motions. Final resting positions show that, in addition to dispersing downslope with travel distances consistent with forms of the generalized Pareto distribution, particles spread laterally along the surface normal to the mean downslope motion (Figure 10). Pooled data for each angularity group reveal both the increased downslope travel distance and lateral dispersion of rounded particles compared to angular particles on slopes from $0.00$ to $0.18$. Final positions for angular particles on slopes $0.25$ and $0.28$ show significant dispersion
across the slope length and width. Several particles on different slopes were visually observed to move laterally away from the mean position and then return closer to this position prior to deposition, leaving the maximum spread unrecorded.

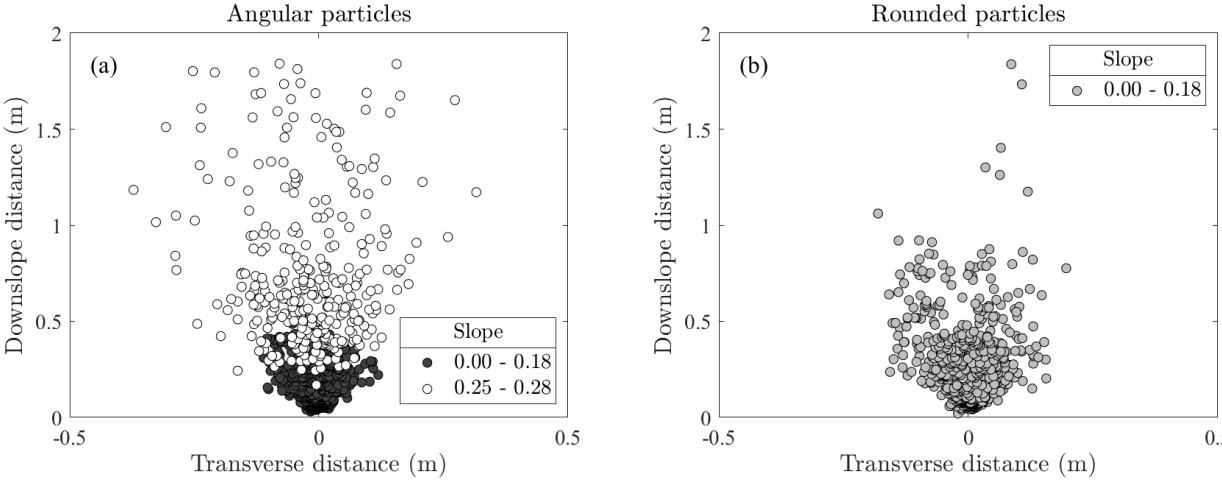

**Figure 10.** Pooled final resting positions of (a) angular and (b) natural rounded particles on the concrete surface of slopes 0.0, 0.09, 0.15, and 0.18. Angular particle positions for slopes 0.25 and 0.28 are depicted in white. The majority of rounded particles did not remain on the experimental surface for slopes greater than 0.18 and angular particle data became highly censored at slopes 0.25 and greater.

By examining the deposition data for individual slopes, we can see the difference in spreading behavior for angular and rounded particles over the range of observed slopes (Figure 11). Mean lateral position is shown in red and does not vary significantly from the center line $y = 0$ of the slope. At $S = 0.00$, rounded particles travel further downslope than angular
particles and experience similar magnitudes of lateral spreading. With a slight increase in slope to $0.15$ the effects of angularity on spreading becomes more apparent. Rounded particles disperse further downslope and experience more lateral spreading. Noticeable transverse spreading is not just observed in the furthest traveling particles, but in the overall population compared to angular particles on the same slope. Note the approximately homoscedastic nature of the data over $x$ for both particle shapes.





Spreading does not appear to be Gaussian with increasing distance downslope, suggesting a more complex partitioning of
energy and likelihood of transverse motion during downslope particle travel than is associated with a Galton board.

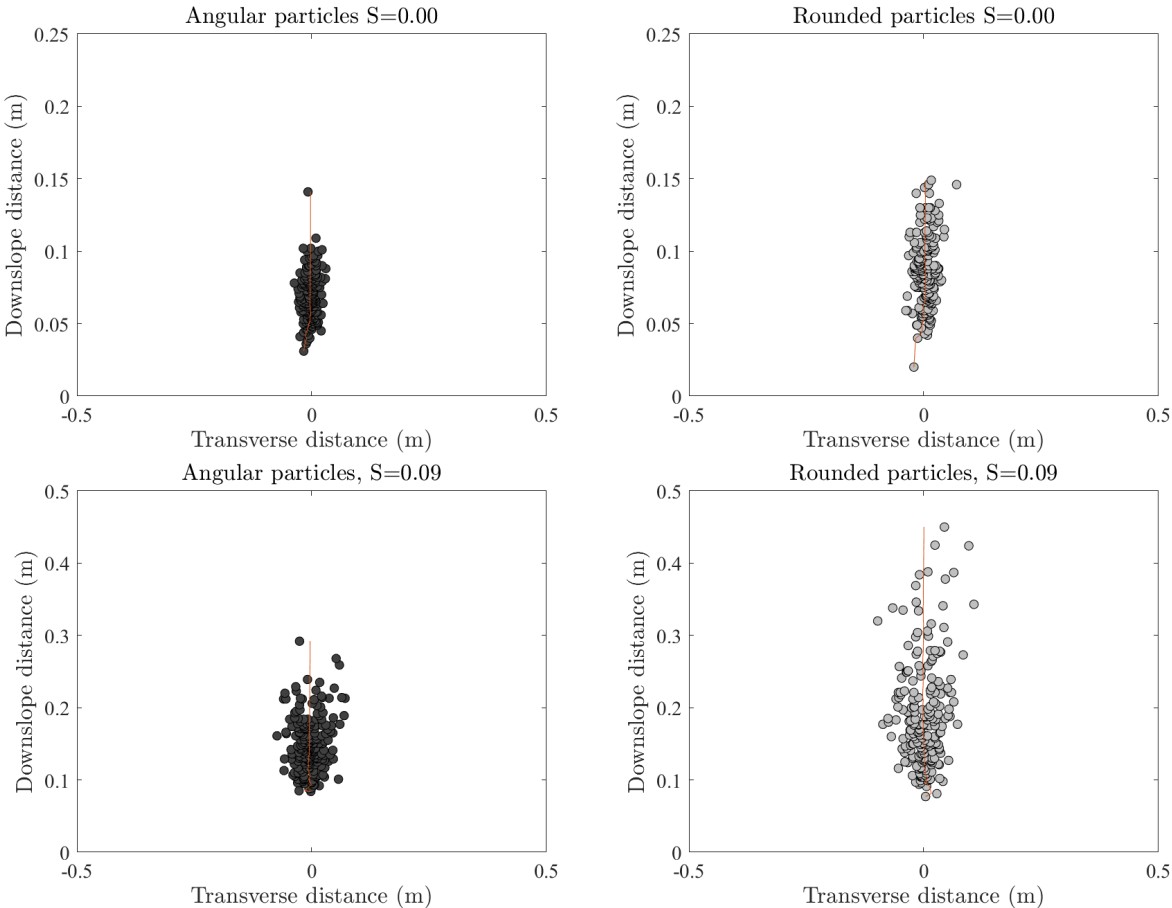

**Figure 11.** Final resting positions of angular (dark grey) and rounded (medium grey) particles on slopes $S = 0.00$ to $S = 0.18$. Cumulative mean transverse positions are indicated in red. Rounded particles disperse further downslope and transversely compared to angular particles. Deposition locations do not indicate late stage spreading like one would expect from Galton board-like behavior.





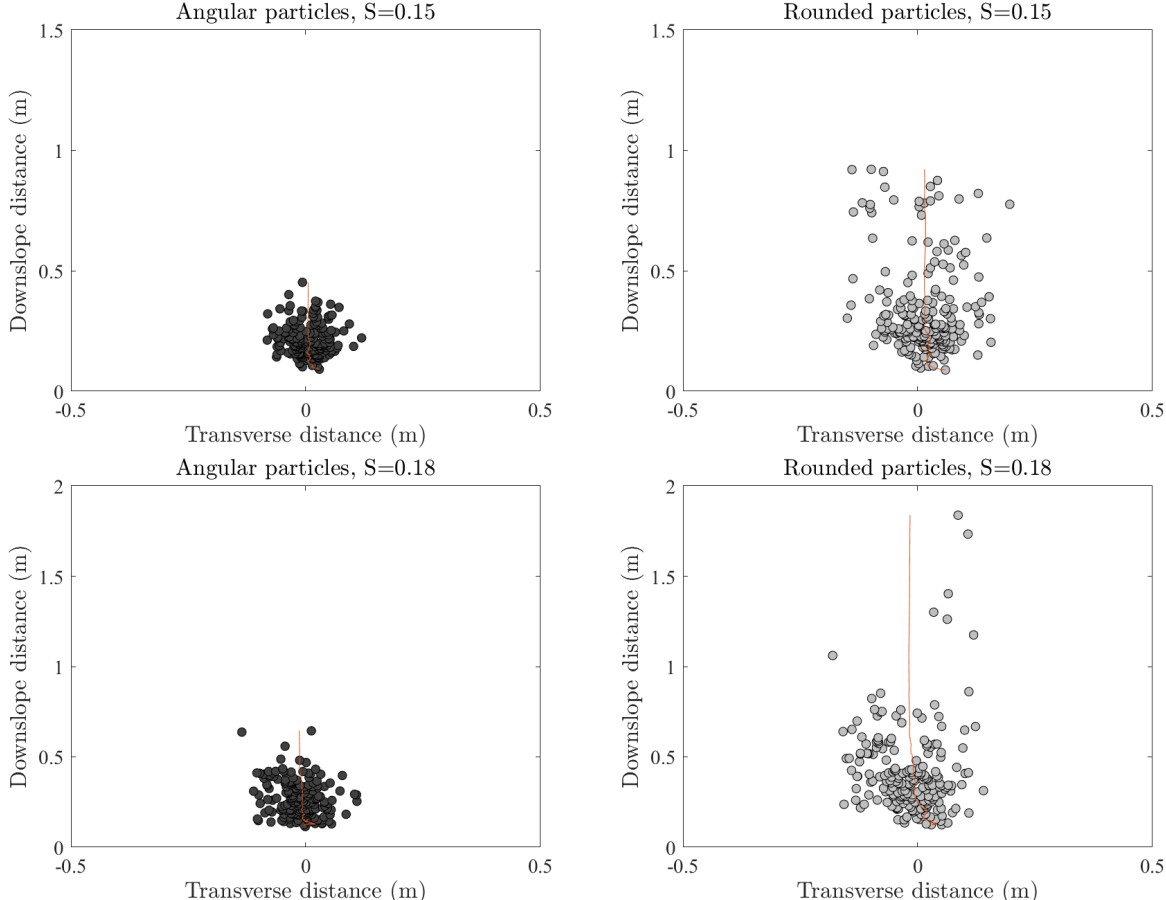

**Figure 11.** Final resting positions of angular (dark grey) and rounded (medium grey) particles on slopes $S = 0.00$ to $S = 0.18$. Cumulative mean transverse positions are indicated in red. Rounded particles disperse further downslope and transversely compared to angular particles. Deposition locations do not indicate late stage spreading like one would expect from Galton board-like behavior.

By visualizing the transverse travel distance for angular and rounded particles using exceedance probability plots, we are able to observe trends in energy extraction with increasing slope in the same manner as before with downslope travel distance. With reference to Figure 12, the concave-down forms of these exceedance probability plots for both particle shapes on slopes less than $S = 0.25$ are akin to thermal collapse as particles spread some finite distance in $y$ before coming to rest. The plots

for angular particles on slopes $S$ equal 0.25 and 0.28 begin to approach nominally isothermal conditions with more linear plots in semi-log space. In contrast to Figure 7, transverse travel distance data do not lead to concave-up plots. Because transverse motion on a surface with zero gradient parallel to $y$ does not directly involve gravitational heating, this motion occurs only due to extraction of downslope kinetic energy or from conversion of rotational energy. Nonetheless, during transverse motion, particles with a downslope component of motion experience gravitational heating. With steeper slopes, this heating increases and



thus provides additional accessible energy for transverse motion (as well as downslope motion). As a consequence, transverse dispersion increases with slope.

Net transverse displacements to the final $y$ position may involve motions in both positive and negative $y$ directions. For this reason we cannot directly appeal to the same ideas of net cooling or net heating, as with downslope motions, to characterize the distribution of net transverse travel distances. On a flat surface with no gravitational heating, the maximum transverse distance is limited by the initial (downslope) particle kinetic energy available for conversion to transverse motion. On a surface sloping parallel to $x$, the maximum transverse distance is limited by the initial kinetic energy plus the accessible kinetic energy gained during downslope motion. Kinetic energy converted to downslope rotational energy is largely unavailable for transverse translational motion. Thus, although we cannot suggest that the linear plots in Figure 12 represent isothermal behavior, we nonetheless suspect that these plots represent a limiting behavior on lateral spreading. These results reinforce the idea that the transverse dispersion-like process fundamentally depends on accessible kinetic energy associated downslope motions, including heating.

To further quantify these interrelated spreading behaviors, we calculate the cumulative variance $\sigma^2_y(x)$ using final $y$ positions about the mean transverse position (modified from Seizilles et al., 2014). This was calculated using

$$\sigma_y^2(x) = \frac{1}{n(x)} \sum_{i=1}^{n(x)} [y_i(x) - \mu_y]^2 \tag{16}$$

where $y_i$ is the lateral position of the $i$th particle, and $\mu_y$ is the cumulative mean lateral position. The total number of particles $n$ ranged from 125 for $S = 0.28$ to 224 for $S = 0.00$. The variances for small values of downslope travel distance $x$ should be viewed with skepticism due to the small sample size used in the calculations. Log-log plots of variance with downslope position are normalized by the average particle diameter $d_p \approx 1$ cm (Figure 13).

The red line with a slope of one is plotted for reference, and the main bodies of data for each slope $S$ are fit with power functions. As described above, transverse spreading as measured by the variance $\sigma_y^2(x)$ increases with slope angle. Because $\sigma_y^2(x)$ denotes a cumulative variance calculated with increasing downslope position $x$, the increase in variance with $x$ in each plot does not represent diffusive behavior as normally envisioned, in which the local variance changes with downslope position. Nonetheless, in all experimental cases spreading occurs at a spatial rate less than that nominally associated with normal spreading behavior, namely, $\sigma_y^2 \sim x$. If associated with Fickian behavior, the cumulative variance would plot with a slope of two.

Both rounded and angular particles spatially exhibit highly sub-diffusive behavior where the fitted lines have slopes less than one. This indicates a change in the effectiveness of conversion of downslope translational energy into lateral motion. Under conditions of net heating, enough energy exists in the system to be partitioned into both downslope and lateral translational motions. Anomalous diffusion may be characteristic of systems such as these, where there is no additional compelling force driving motion other than gravity. On a hillslope, there is a fixed amount of potential energy that can be partitioned before the particle reaches the bottom of the slope. Particles have maximum potential energy available to be converted to kinetic energy at the crest of the hillslope and have a maximum kinetic energy at some point between this initial position and deposition location.



Earth **Surface**
**Dynamics**
Discussions

EGU

**Figure 12.** Plots of exceedance probabilities versus the absolute value of transverse slope travel distance for experiments over six values of slope $S$. The transverse mean position remains close to zero demonstrating that there is no preferential drift to $-y$ or $+y$ as illustrated in Figure 11. The concavity of the plots suggest that transverse dispersion is dominated by cooling. No energy in addition to that partitioned from downslope motion is put into lateral spreading.





**Figure 13.** Transverse particle position variance $\sigma^2$ of angular (dark grey) and rounded (medium grey) as a function of downslope position $x$ plotted in log-log space. Values are normalized by the average particle diameter (1cm). A line of slope 1 is plotted as a reference. The main bodies of data plotted are fit with power-functions depicted in red to illustrate the difference in diffusion behavior. Tails of the data plotted are not included in the fit as these variance values are distorted by low sample numbers, at small downslope positions, or truncation of data, at larger downslope positions.





Energy may be lost immediately due to collision with the surface resulting in rotational motion downslope and/or laterally,
translational motion downslope and/or laterally, or energy lost to some other form. As the effectiveness of gravitational heating
increases with slope, the effectiveness of frictional cooling remains approximately constant. Particles that lose energy immediately are unlikely to travel far in either direction, which is apparent in the magnitude of lateral spread of angular particles
on high slopes compared to rounded particles (Figure 11). At a later stage, normal spreading is not observed even at high
slopes based on the flattening variance curves with increased downslope distance (Figure 13). It is important to note that the
data presented here are composed of deposition coordinates, which, while not unusual for hillslope studies (Schumm, 1967;
Gabet and Mendoza, 2012), do not allow for similar analysis as in high-speed imaging of motions in flume experiments. Final two-dimensional travel distance data, however, provides an important step in incorporating the transverse direction into
formulations of sediment motion and sediment flux on hillslope surfaces (see next section).

## 6  Discussion

The laboratory experiments presented above demonstrate that particle motions are distinctly probabilistic in behavior due
to inherent variability in energy extracted when moving over a surface. Particle properties, especially angularity, influences
particle travel dynamics in a top-down manner. The angularity of particles directly affects the energy extraction during particle-
surface collisions, and in turn the distances that particles travel downslope and in the transverse direction. Even on a relatively
smooth surface, angular particles lose energy to friction or rotation more readily than spheres or comparably sized rounded
particles. Surface characteristics, such as relative surface roughness, also influence travel dynamics but in a bottom-up manner,
as in the manner of a Galton board. Angularity, in addition to surface roughness, introduces randomization of motions and is
thus an important element of the transport problem. An important, open question concerns the extent to which top-down effects
of particle shape (rounded versus angular) on energy extraction and travel distances are discernible from those associated with
a bottom-up control due to surface roughness.

Lateral particle dispersion is a key component of transport in many settings and a better understanding of the observable
kinematics and underlying mechanics is needed to fully describe sediment transport. Transverse spreading, though neglected
in hillslope literature, has been described in a few fluvial bedload transport projects but analysis is largely limited to kinematic
interpretation (Lajeunesse et al., 2010; Roseberry et al., 2012; Seizilles et al., 2014). The analysis offered here are starting
points for conceptualizing sediment transport on hillslopes as a two-dimensional problem involving lateral dispersion. In the
same way that sediment may be delivered at some position $x$ at differing rates due to upslope conditions, sediment delivery
is also influenced by transverse slope conditions due to lateral motions of many particles over time. The importance of two-
dimensional travel of particles is readily apparent during many processes. For example, with rain splash transport on a flat
surfaces, particle displacement is symmetrical in all directions about the drop impact (Furbish et al., 2007). The average net
displacement of particles in the presence of finite surface slope yields an approximate slope dependent transport rate whose
effect is to contribute to "diffusive smoothing" of irregular surfaces. But in addition, over many raindrop events smoothing
may occur due to the two-dimensional radial displacements of particles whose effect is to evenly spread the particles. Lateral





particle dispersion during hops in the presence of downslope advection similarly results in increased surface smoothing of irregular surfaces. This behavior offers a simple way of conceptualizing the need to characterize the flux in two dimensions (Appendix A).

Consider an idealized cliff that supplies particles as a line source to a hillslope with transverse variations in elevation reminiscent of swales or gullies formed by transverse variations in particles delivery rates or perhaps formed by channelized flow (Figure 14).

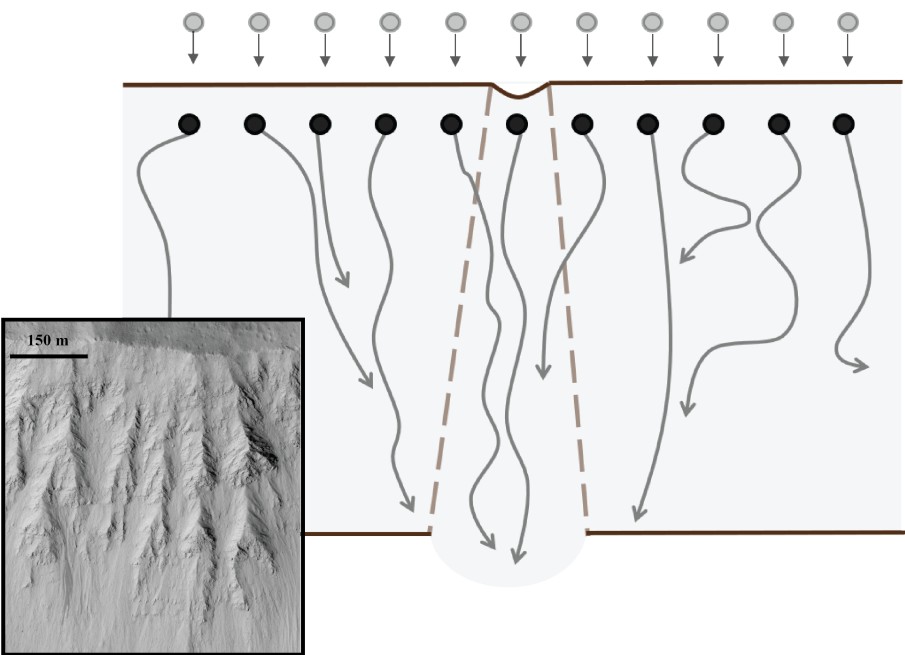

**Figure 14.** Conceptual illustration of lateral dispersion of a line of particles delivered to a slope with a gully-like indentation. Transverse spreading of particles as they travel down slope result in filling of an the indentation leading to surface smoothing. Inset is a rendering of eroded crater with gullies in HiRISE image PSP_007456_2140 as an example of smoothing in landscapes due to particle diffusion. Original Image credit: NASA/JPL/Univesity of Arizona

Over time many particles are delivered to the hillslope and travel downslope with travel distances described by, for example, a generalized Pareto distribution. As the particles interact with the surface, they spread in the cross-slope direction due to energy

conversion into transverse translational kinetic energy. The particles spread relative to the initial width over which that they were delivered to the hillslope. Particles that fall into the swales are likely to remain within the depressions. Particles shed onto the slope, but not initially channelized into a swale, continue their downslope motions on the surface. Some particles will deposit on the elevated surface, while some will spread laterally into the neighboring swale. After some period of time, enough sediment is delivered to the surface and undergoes spreading such that the depressions on the surface become filled.

As swales are filled upslope by sediment, the surface acquires a more uniform slope at all lateral positions for the same downslope coordinate. Sediment diffusion in the cross-slope direction thus acts to smooth topographic roughness, which is



less likely with purely one-dimensional transport (Schumer et al., 2009). Thus, forms of natural hillslopes may indicate the competition between multi-dimensional roughening and smoothing processes as these impact particle flux and land-surface elevation change (Appendix A).

**7 Conclusions**

1. Particle shape plays a dominant role in how energy is partitioned during impact with a surface. Relative to spherical and natural rounded particles, angular particles give greater variability in rebound behavior.

2. The effects of particle shape on energy conversion is especially pronounced when considering multiple collision events during downslope travel. Angular particles travel shorter distances than rounded particles for the same surface slope
although downslope travel distance data for both angularity groups are well fit by generalized Pareto distributions.

3. Consecutive particle-surface collisions during downslope travel lead to transverse particle dispersion, the magnitude of which depend on particle shape and surface slope. Transverse particle diffusion during downslope motion may contribute to a cross-slope particle flux, and likely contributes to topographic smoothing of irregular hillslope surfaces.

4. This random-walk behavior represents a top-down control on the randomization of particle trajectories due to particle
shape, which is in contrast to a bottom-up control on randomization of particle trajectories associated with surface topography. Surface roughness is not the only factor that influences downslope and transverse travel distances, at least in rarefied systems.

*Data availability.* The data and supporting information presented here may be accessed via the Vanderbilt University Institutional Repository (http://hdl.handle.net/1803/9742).

**Appendix A: Two-dimensional downslope flux**

For completeness with reference to future work, here we provide the two-dimensional versions of the entrainment forms of the particle flux and the Exner equation. The latter has been described previously in relation to bed load transport (Furbish et al, 2012). Only the one-dimensional version of the entrainment form of the flux has been previously presented (Furbish et al., 2012b; Furbish and Roering, 2013; Furbish et al., 2017). Our objective is to reinforce points in the text as illustrated in Figure
14, that transverse spreading may give a cross-slope flux, and that both the downslope flux and the rate of change in the surface elevation at a position $(x, y)$ may involve particles whose motions started upslope at lateral positions different from $y$.

Let $r$ ($0 \leq r < \infty$) denote the downslope particle travel distance parallel to $x$ and $s$ ($-\infty < s < \infty$) denote the transverse particle travel distance parallel to $y$. The joint probability density function $f_{r,s}(r, s; x, y)$ of the travel distances $r$ and $s$ for



particles whose motions start at position $(x, y)$. Let us instead define an exceedance probability function as

$$R_{r,s}(r, s; x, y) = \int\limits_{s}^{\infty} \int\limits_{r}^{\infty} f_{r,s}(r', s'; x, y)\, \mathrm{d}r'\, \mathrm{d}s'. \tag{A1}$$

This is the probability that the downslope travel distance is greater than $r$ *and* the transverse travel distance is greater then $s$.

Consider the position $(x, y)$ on a hillslope, and let $\Delta y$ denote a small interval starting at $y$. We wish to calculate the downslope particle flux through this interval. Let $x'$ and $y'$ denote starting coordinates. For a particle starting at position $(x', y')$ to move downslope through the interval $\Delta y$ at position $(x, y)$ it must travel a downslope distance greater than $r = x - x'$ and a transverse distance greater then $s = y - y'$ but less than $s = y + \Delta y - y'$.

The probability that a particle beginning motion at $(x', y')$ will travel a downslope distance greater than $r = x - x'$ and a transverse distance greater than $s = y - y'$ is

$$R_{r,s}(x - x', y - y'; x', y'). \tag{A2}$$

Similarly, the probability that a particle starting from $(x', y')$ will travel a downslope distance greater than $r = x - x'$ and a transverse distance greater than $s = y + \Delta y - y'$ is

$$R_{r,s}(x - x', y + \Delta y - y'; x', y'). \tag{A3}$$

Therefore the probability that the particle will travel downslope through the interval $\Delta y$ is

$$R_{r,s}(x - x', y - y'; x', y')$$

$$-R_{r,s}(x - x', y + \Delta y - y'; x', y'). \tag{A4}$$

The next task is to integrate over all possible starting positions $x'$ and $y'$, incorporating the entrainment rate $E(x', y')$. This integration gives

$$\int\limits_{-\infty}^{\infty} \int\limits_{-\infty}^{x} E(x', y') R_{r,s}(x - x', y - y'; x', y')\, \mathrm{d}x'\, \mathrm{d}y'$$

$$-\int\limits_{-\infty}^{\infty} \int\limits_{-\infty}^{x} E(x', y') R_{r,s}(x - x', y + \Delta y - y'; x', y')\mathrm{d}x'\, \mathrm{d}y'. \tag{A5}$$

If the entrainment rate $E$ denotes a volume of sediment per unit area per unit time, then Eq. (A5) represent a volume of sediment per unit time. If $E$ denotes a number of particles per unit area per unit time, then Eq. (A5) is a number of particles per unit time. To make this a flux we divide by the interval $\Delta y$ to give





$$q_x(x,y;\Delta y) =$$


$$\frac{1}{\Delta y}\Bigg[\int\limits_{-\infty}^{\infty}\int\limits_{-\infty}^{x} E(x',y')R_{r,s}(x-x',y-y';x',y')\,\mathrm{d}x'\,\mathrm{d}y'$$

$$-\int\limits_{-\infty}^{\infty}\int\limits_{-\infty}^{x} E(x',y')R_{r,s}(x-x',y+\Delta y-y';x',y')\mathrm{d}x'\,\mathrm{d}y'\Bigg]. \tag{A6}$$

The formulation shows that in order to calculate the downslope flux through an interval $\Delta y$ at position $(x,y)$, one must

consider transverse motions that deliver particles to this position. Here it is important to note that, whereas the exceedance probability function $R_{r,s}(r,s;x,y)$ may not vary with $y$, variations in the entrainment rate $E(x,y)$ over $y$ can contribute to variations in the downslope flux $q_x(x,y;\Delta y)$.

In turn, let us consider the local elevation of a surface denoted as $\eta(x,y)$. The entrainment form of the Exner equation in two dimensions is

$$c_b\frac{\partial\eta(x,y)}{\partial t} = -E(x,y)+$$

$$\int\limits_{-\infty}^{\infty}\int\limits_{-\infty}^{x} E(x',y')f_r,s(x-x',y-y';x',y')\,\mathrm{d}x'dy' \tag{A7}$$

where $c_b$ is the volumetric particle concentration of the sediment surface and $E(x,y)$ is the volumetric entrainment rate. The disentrainment rate function, defined as

$$P_{r,s}(r,s;x,y) = \frac{f_{r,s}(r,s;x,y)}{R_{r,s}(r,s;x,y)}, \tag{A8}$$

is vital to connecting descriptions of flux and divergence to the mechanics of particle motions and disentrainment. Thus, the distribution $f_{r,s}(r,s;x,y)$ and its survival function $R_{r,s}(r,s;x,y)$ in Eq.(A7) and Eq. (A6) are central to descriptions of the volumetric flux of sediment particles and the associated surface elevation change. Time variations associated with ensemble expected values for downslope flux $q_x(x,y)$ and land-surface elevation $\eta(x,y)$ (Furbish and Haff, 2010) are not included in

these formulations.

*Author contributions.* The reported work represents an intellectual co-conspiracy between the authors. Writing represents the work of SGW with contributions from DJF.

*Competing interests.* We have no competing interests.



*Acknowledgements.* We acknowledge support by the U.S. National Science Foundation (EAR-1420831 and EAR-1735992). We greatly

appreciate Brandt Gibson for helping us set up the experimental slope. Many thanks to Rachel Glade for her preliminary thoughts on lateral

motion, and to Kristen Fauria and Shawn Chartrant for their comments during editing.





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
