# Peer review of "Particle energy partitioning and transverse diffusion during rarefied travel on an experimental hillslope"

_Earth Surface Dynamics, 2020_

## Author Comment (AC1)

We greatly appreciate the comments and suggestions of the two referees for this work. Our revisions in the manuscript and responses below are provided in blue font.

We note that the reviews for this paper are primarily focused on the need to provide context relative to previous work and to simplify the presentation of materials. In response, we have added numerous clarifying statements and removed several unnecessary passages. We have, however, made no changes to the technical elements of this work and have opted to retain much of our original stylistic choices.

**Reviewer 1**

Authors investigate particle motions when collide on rough hillslope surfaces; in particular the contribution of kinetic, rotational, and frictional energies during their travel going downslope using recent works from Furbish et al. (2020). They analyzed from very interesting experiments the influence of the geometry particles on the energy conversion. The transverse spreading during the travel of particles on different rough surfaces is discussed.

1. Concerning the experiments: Could you describe more precisely what you call "a smooth slate surface" ?

   The surface used is a machined slate that is smooth but not polished, akin to a slate tabletop that has not been finished. **We have added a parenthetical statement to clarify this point.**

2. Many authors suggest (Brach included) that a well determination of the coefficient of restitution, the energy relating to all six degrees of freedom (translational and rotational) should be obtained. The interesting discussion about the influence of the particle shape in the present work would be more fruitful if you discuss on the base of tangential and longitudinal coefficient of distribution and the collisions depending on the shape of the particle. I guess that you have all the data (after seeing your supplementary materials) to estimate tangential and longitudinal coefficients of restitution, which are related to the contact forces during the impact and then, the partition between energies. See for example, S. Dippel, G. G. Batrouni, D. E. Wolf, PRE 1997 and 1998; M. Louge et al 10.1103/PhysRevE.65.021303; Higham https://doi.org/10.1007/s10035-019-0871-0, and references therein. I think that authors con...

   Based on our experimental set-up with only one camera, we cannot resolve elements of motion relative to the plane of the particle trajectory following collision, making such a thorough analysis impossible. We note that Higham et al. (2019) had several cameras and could therefore extract this kind of information. We agree it would be an interesting discussion, but for the objectives of our work (including applications to complex natural hillslopes) we prefer to simply demonstrate that the proportion of energy lost during collision can be described as a random variable that is strongly influenced by particle angularity. With Figure 5 we do suggest, although qualitatively, that there are variations in the degrees of freedom for collisions of particles with increasing angularity.

   **We have added phrases of clarification — one in section 4.1 explaining the limi-**

**tations of our single camera set-up and one in the paragraph preceding Figure 5 that notes the uniqueness of these coefficients of restitution.**

3. Downslope distance

   Henrique et al (PRE 1998 , 57, 4) report on an experimental, numerical, and theoretical study of the motion of a ball on a rough inclined surface. Among the control parameters, the initial kinetic energy is one of them. The authors analyze the dependence of the traveled distances on and the kinds of mechanism of dissipation depending on the initial kinetic energy which is a constant friction force or viscous for small initial energies. They showed that a ball that has a large enough initial kinetic energy first bounces on the rough surface and suffers a constant friction force and the ball could not be trapped if its velocity is larger than the crossover velocity. After this threshold, the friction force suddenly becomes viscous. In fact, they show the existence of two mechanisms of dissipation, i.e., friction forces, related to the difference in the nature of the collisions when the energy is below or above this threshold.

   (a) How do you compare your results to those of Henrique et al in the particular case of rounded grains?

   We first note that the velocities of the particles during motion in our experiments were not recorded due to limitations in our experimental set-up. Several video recordings of particle launching were taken only to estimate initial kinetic energy. Instead, we focused on particle final resting positions for these experiments.

   The initial kinetic energy was varied with slope based on our experimental set-up, so we are unable to comment on the independent effects of each on particle motions in the same way as Henrique et al. (1998). Our results do show an increased travel distance with increased slope (or initial kinetic energy) in the same manner as the previous experiments, which is expected. However this increase in travel distance is more related to gravitational heating than to initial energy. The addition of degrees of freedom (i.e. complexity of collisions) introduced by the particle angularity in our experiments, even for the rounded particles, makes a direct comparison of our results to those of Henrique et al. difficult. We do not have the data to support the idea that friction is either constant or viscous, as Henrique et al. (1998) suggest. We observed that particles, both rounded and angular, experience a collisional form of friction throughout travel, whereas the use of spheres in the referenced experiments allowed for Coulomb-like or viscous-like friction to dominate after a certain amount of energy loss. The role of angularity on energy loss in this sense is clear, as particle angularity effectively precludes a Coulomb-like regime. We emphasize that the tumbling motions, even for the rounded particles, lead to a dominantly collisional friction. Figure 8 further reinforces this point, as the data collapse to a collisional friction model offered in Furbish et al. (2021a, b). **We have added clarity to the paper near Figure 8 to reinforce this point.**

   (b) What you can say about the stopping distance when grains are trapped by the surface?

   We first offer the reminder that our surface is relatively smooth with only a granular sand-scale roughness that is much smaller than that of our gravel-sized particles.

This means that there are effectively no traps, as described by Henrique et al. (1998), on our experimental surface. Stopping in these experiments occurred where gravitational heating outweighed cooling from collisional friction, and these factors are in part simultaneously determined by particle angularity and slope angle.

Surface roughness is an additional bottom-up controlling factor on energy dissipation but is likely less important at high slope angles. With higher slopes and higher initial kinetic energies, the increased particle momenta relative to those of particles on lower slopes likely result in less interaction with the surface and thus less of an influence by surface roughness on particle motions. Collisional events which dissipate energy may be less frequent or the amount of energy dissipated may be smaller relative to the available energy such that travel is less influenced by surface roughness. **We have added a statement to clarify that large roughness traps capable of stopping the particles are absent from the surface.**

(c) Did you find a threshold when varying the slope? and with the initial kinetic energy?

As a point of reference, the formulation (Furbish et al., 2021a) underpinning our analysis does not appeal to differences in frictional behavior, and no thresholds are involved. That said, the crossing of isothermal conditions in transition between a bounded and heavy-tailed distribution could be roughly interpreted as analogous to the transition between frictional regimes leading to changes in sphere behavior as presented in Henrique et al. (1998). Nonetheless, we reiterate that this does not represent a threshold associated with a change in frictional behavior — a point reinforced by the collapsed plot of travel distances in Figure 8, which is based on the occurrence of collisional friction for both bounded and heavy-tailed forms of the distribution.

(d) Are the difference between the travel distances for angular and rounded particles launched on the roughened experimental surfaces due to different mechanisms of dissipation? Which are the origin of those mechanisms? Are only due to the impact?

Unlike the rolling sphere on a monolayer roughness, which experiences viscous-like or Coulomb-like friction, the motions observed in these experiments are best described as tumbling where the friction regime is entirely collisional. The differences in travel distance are not due to different mechanisms of energy dissipation but rather due to the angularity of each particle. Collisional friction dictates the dissipation of energy for both rounded and angular particles, although the effects appear more pronounced for angular particles. This is directly reflected in the drop experiments. During travel, particles may appear to be rolling. On closer examination, however, we see that the motions are better described as tumbling and the frictional regime remains collisional due to particle angularity, where the particles do not maintain contact with the surface during tumbling. We choose to define energy dissipation in our experiments as any change in energy from that contributing to downslope translational displacement. This includes energy partitioned into particle rotation, translation into the lateral directions, or other forms of energy dissipation. Particle angularity not only influences collisional geometry which dictates

energy lost with collision, but it also makes it unlikely that particles experience rolling motions like the spherical particles in the experiments of Henrique et al.

We suggest Appendix J from Furbish et al. (2021a) to further elaborate this point.

**We have added a statement clarifying that both rounded and angular particles experience collisional friction and motion is best described as tumbling rather than rolling.**

4. Lateral spreading

   (a) Authors said that they measured lateral position from experiments performed above. Did you vary the initial energy at which was launched each grain? It is known that in a system like Galton billiard model or random walk, particles, after several collisions, they lost their memory. Did you explore that typical distance for your grains of different geometries? Or the number of collisions? (Which is more difficult to measure in this case).

   Initial energy was varied proportionally to slope as the two were interconnected by experimental set-up. The formalism presented in Furbish et al. (2021a) suggests that initial energy is a component of the problem, however the influence of collisional friction on a particle quickly surpasses initial momentum of the particles such that the effects of initial conditions rapidly vanish. We do not have data support this claim further as only the final resting position in both the downslope and lateral directions were recorded. We qualitatively observed that some particles traveled max distances in the lateral direction prior to moving back towards the center. This behavior indicates this loss of memory as mentioned, as observed max lateral travel distances did not necessarily occur at the final deposition location.

   To answer the second part of the question, we did not explore the number of collisions although we observed that the number of collisions increased with downslope travel distance. We are unsure whether rounded versus angular particles interacted with the surface more or less frequently and how this may change with increasing slope. Upcoming experiments will be in part aimed at better quantifying how frequency of collision changes over space and time in relation to travel distances

   **We have added a statement concerning the rapid loss of particle memory due to the influence of collisions.**

   (b) The lateral variance indicates that when you increase the slope a constant value is reached with the downslope position which varies between $10d_p$ or $20d_p$ from their plots. It is not easy to find typical values in some curves over 1 decade. Did you have any explanation about those typical lengths? Those typical lengths are quite related to some velocity correlation length that you will serve to characterize the difference between rounded and angular grains and their random motion.

This work focused on particle final resting position and in so we do not have the dynamic information needed to determine velocity correlation length scales. Figure 13 is displaying cumulative variance plots associated with final resting positions which are distinct from dynamic plots showing variance growth in time or space that have been used in previous treatments of the problem. Based on Figure 10, it appears that the local number of disentrained particles (per downslope distance) decreases with increasing downslope distance (especially for $S = 0.25$ and $S = 0.28$) in approaching the longest recorded travel distances. This gives a decreasing relative contribution to the calculated cumulative variance at large travel distances. In this context, it appears that the flattening of the slopes may not have a physical explanation, but rather represents a statistical artifact.

I think that it is a very nice and interesting work that can be published after revision. In particular, authors need to discuss comparing the present study to previous works, as I said before, in order to provide physical arguments for the mechanisms involved and observe by the experiments.

**Reviewer 2**

The authors present high-speed camera measurements of rounded and angular grains interacting with a rough surface in drop-rebound and downslope transport experiments. Their major findings are that (1) grain-surface collisions with angular grains convert more gravitational energy to rotational than do collisions with more rounded grains, leading to generally shorter travel distances; (2) the transverse rate of spreading is dependent on the available downslope translational energy; and as a consequence, (3) the rate of transverse spreading is contingent on grain shape as well as surface roughness, as more angular grains have generally less downslope momentum available for conversion to cross-slope momentum due to their more rapid loss of gravitational energy into particle rotation.

The paper is good and I see no major issues with the scientific ideas. The presentation is mostly easy to follow, the experiments are well-designed and thoroughly explained, and the analyses in the paper easily convince the reader of its main conclusions. Although these conclusions may seem rather specific to grain-scale sediment transport processes, the authors do a nice job of explaining the wider implications for hillslope evolution, so the work should be of broad interest to many ESurf readers. However the writing includes some perhaps unnecessary terminology, some variables and concepts are explained at length without ever being directly linked to the rest of the paper, certain explanations intended to provide intuition can be rather slow moving, and several figures could be improved to better explain the experimental configuration. I therefore recommend publication with minor revisions, as I explain below for the authors.

**Main suggestions:**

**A key issue is that a lot of background info you included in the paper (Secs. 1-3 and appendix A) does not directly contribute to the central findings of the paper and is readily available elsewhere.** For example, Eq 1. and appendix A describe the evolution of topography under sediment transport, but you never measure or calculate elevation change. Similarly Eq 2. represents the sediment flux, but you never measure or calculate a sediment flux. Eq. 7 introduces the length scale of deposition, but this quantity is never mentioned again. The exceedance probability Eq. 13 is used repeatedly, but the non-exceedance probability Eq. 12 is trivially related, available elsewhere, and is not used once. The text at L175 seems to constitute "relevant theory" suggesting it belongs instead in section 3. The presentation of the Furbish et al (2020a) model for the generalized Pareto distribution in section (3) is described carefully but this material could be briefly outlined instead, as this theory is available in Furbish et al (2020a). Finally, much of the discussion setting up a conceptual particle cohort between L110 and L125 could easily be cut without causing any confusion.

> We intend for this paper to "stand alone" with limited, albeit perhaps necessary, reference to the preceding papers. We therefore have purposely included these background elements to motivate the problem, and will retain much of the introductory material. Because we are more broadly interested in descriptions of the particle flux and its divergence, we focus here on understanding the distribution of hop distances and associated exceedance probabilities. We view Eq. (1), Appendix A, and Eq. (2) as being key in motivating the problem, even without further reference to them in the paper. We retain Eq. (7) because it indicates that the deposition length scale containing $\beta_x$, as described by Eq. (6), does not involve Coulomb-like friction. This reinforces the point that the disentrainment processes described here are collisional in nature. We have, however, trimmed Eq. (12) as suggested.
>
> **In the text we have added a statement to clarify this point, in particular that Eq. (1) and Eq. (2) connect the kinematics of particle motions to the underlying physics of disentrainment.**

Given these observations, I suggest **to reorient the material in section 3, 4, and Appendix A toward the main stream of the paper.** One way to do this is to cut Eq 1, Appendix A, Eq 2, and Eq 12, paraphrase L110-125, and incorporate the paragraph at L175 into Section 3. Perhaps given the reference to the ideas underlying the 2D Exner equation near L444 and your desire to include it, you could state the equation there at L444 without a derivation by referencing Paola and Voller (2005), then use its structure to support your discussion about topographic smoothing. Finally, since you analyze two dimensions, I suggest to make Figure 1 two-dimensional, so it shows diffusing down-slope particles, collisions without friction, the coordinate system, and the definition of $\beta_x$ and $\beta_y$ if possible. This would help the reader to clearly understand all elements within your experiments. For brevity, you might consider combining Figure 1 and Figure 6 in a two-panel figure, showing the experimental setup with the particle launcher as an inset in one panel, and the conceptual diagram in the second. This is just one suggested set of revisions to focus the introduction of the paper toward the problem at hand. However you choose to do this, the paper would certainly be easier to follow after some effort to focus the introduction (mainly in section

3) to fit the paper's narrative, highlight the additional spatial dimensions of hillslope sediment transport you've analyzed, and define important variables in a modified Fig. 1 concept sketch.

> We appreciate the interest and suggestions, but we prefer our stylistic interpretations and the presentation of this paper as a standalone work. Additionally, we have chosen to present the two-dimensional versions of the flux and entrainment form of the Exner equation to address the point of topographic smoothing addressed in the discussion and conclusions. We note that the formulation presented by Paola and Voller (2005) is built entirely on a continuum framework and has little to do with the framework and formulation presented in this paper. To suggest that we state the equations from Appendix A without derivation and only with reference to Paola and Voller (2005) does not make sense.

> **We have created a 2D version of Figure 1 and further clarified the caption.** The suggestion of combining Figures 1 and 6 is interesting as well, but we worry that this may lead to confusion and we prefer to keep them separate. Figure 1 is a general definition diagram; and because the particle drop experiments occur in the presentation before the travel distance experiments, it is premature to show the travel distance experimental set-up so early in the introductory material, as Figure 1 must stay in Section 2.

**Minor comments:**

L1: The abstract is nice. However, self-citations of recent papers in the first line of the abstract to support well-known ideas runs the risk of appearing egotistical. I suggest "Particle motions down rough hillslope surfaces act to balance energy supplied by gravity against energy dissipated by collisions" (or similar) so the authors claim less ownership of these well-established ideas (from Kirkby or earlier).

> **We have removed the first part of the first sentence.** The rest of this sentence is accurate and only recently clarified, noting that the formulation of Kirkby and Statham (1975) and others involved a Coulomb-like friction rather than collisional friction, attributable to Riguidel et al. (1994), Sampson et al. (1999) and Quartier et al. (2000), then advanced by Gabet and Mendoza (2012) and formalized in Furbish et al. (2021a, 2021b). See also our related responses below concerning the ideas of particle acceleration and deceleration.

L28: Isn't it Einstein 1937, not '38?

> Corrected

L32: Do you mean to say "with the associated mechanics of particle *transport*"? – are you discussing disentrainment in particular? Or do you mean to say that the entrainment rate and the particle travel distance distribution leads to the deposition rate?

> We are discussing the process of disentrainment in particular.

L41: What is your distinction between disentrainment and deposition? This is individual grains vs many grains? If so, is the distinction used consistently?

Disentrainment is a probabilistic (mechanical) concept. Indeed, the disentrainment rate is a probability per unit distance, and in essence represents a spatial Poisson rate constant with dimension $[\mathrm{L}^{-1}]$. Deposition, on the other hand, is intended to refer to its ordinary meaning of coming to rest, although this also is implied by disentrainment. **We have added this clarification to the text.**

L50: Suggest "summarize the relevant theory" with regard to the long text I wrote above about reorienting the intro toward the objectives

Note previous response.

L79: Regarding the "heating" and "cooling" terminology, as mentioned on the reviews of some other recent ESurf papers by the authors, it is certainly not consistent with temperature concepts from gas theories, which define temperature as the velocity fluctuations away from the ensemble mean velocites, not the ensemble mean velocities themselves (as in the present context). It's much like calling a second moment of position a variance, which is wrong. The terminology does not cause conflicts at this stage, but we have to wonder if it will as granular gas ideas become more integrated in the grain-scale sediment transport theory which you are advancing here. The more standard acceleration/deceleration terminology poses no such issues as far as I can tell, and in fact it makes ideas more clear (to me). For example consider a modified Fig 2 caption: "(a) A<0 representing collision-dominated deceleration and (b) A>= 0 ... representing gravity-dominated acceleration." What about this well-established (probably hundreds of years old) terminology needs reworking?

Concerning the last two sentences above, we are not reworking well established terminology, and this interpretation of the shape parameter $A$ is incorrect. The formulation does not (and cannot) simply appeal to the concepts of acceleration and deceleration. Due to the probabilistic nature of the problem, transport and disentrainment of a cohort of particles can involve net heating $(\mathrm{d}E_a/\mathrm{d}x > 0, A > 0)$ without an overall (average) acceleration due to preferential culling of lower energy particles during deposition. This point is elaborated in our response to a question below, where we note that associating net cooling $(A < 0)$ with deceleration and net heating $(A > 0)$ with acceleration is incorrect when describing the ensemble behavior of particles.

We fully address the other items offered above in our response to the reviewer comments on Furbish et al. (2021a) (https://esurf.copernicus.org/preprints/esurf-2020-98/). Here we summarize key points from that response. The formulation of particle transport and disentrainment under rarefied conditions certainly appeals to concepts from kinetic theory of granular gases. However, this transport problem cannot be viewed or treated as one might an ordinary granular gas. The rarefied conditions described here only involve particle-surface collisions, not particle-particle collisions. As such, the Knudsen number is effectively infinite. Essentially all collisions extract particle energy associated with downslope motion, whereas particle-particle collisions in an ordinary granular gas can locally add energy to the particles. (Indeed, this is why a dissipative granular gas exhibits a stationary Maxwell-Boltzmann-like velocity distribution, albeit with exponential tail decay, in the homogeneously driven condition and in the homogeneous cooling state.) Addition of energy (heating) in our problem only involves gravity, not collisions. Whereas one can formally define a granular temperature at a position $x$,

this temperature is not physically meaningful, and its behavior certainly could not be associated with, say, Haff's cooling law. (The formulation in fact does not appeal to a granular temperature.) The particles do not possess an "internal" energy and a granular pressure does not exist. The particle system does not evolve dynamically in time as does an ordinary granular gas. That is, there are no internal gas dynamics due to particle-particle interactions, and granular energy is neither advected nor diffused over space. Because of these fundamental differences we are not compelled to unnecessarily match our description with kinetic theory of ordinary granular gases. And, as noted above (and elaborated below), the suggested acceleration/deceleration idea misses the mark. In short, as described in Furbish and Doane (2021):

> "The objective therefore is to aim at probabilistic descriptions of sediment particle motions and transport that lean on the *style of thinking* of statistical mechanics, recognizing that this endeavor is not simply about adopting established theory or methods "off the shelf." Rather, such efforts involve tailoring descriptions of transport to the process, the scales of interest and the techniques of observation and measurement used."

We therefore retain our description of particle energy involving heating and cooling during transport and disentrainment, without reference to granular temperature.

Fig 2 caption: "Plot of..." is not needed unless you really want it. It's clearly a plot of something! This same comment applies to many figure captions in the paper.

We prefer to include such phrases to better describe the type of figure being displayed for those potentially relying on reading software.

L180: This is not exactly accurate. Plenty of granular gas theory studies consider the restitution coefficient a random variable, whether directly or by parameterizing it by the particle velocity. See for example Serero et al (2015).

Having recognized this point during the review process for Furbish et al. (2021a), we have modified the text appropriately.

L185: Gravel-sized ?

Corrected

L194: 20202020

Corrected

Figure 4. This is a nice plot! I see that you fit the data to the CDFs. I am curious though why you choose not to indicate the empirical frequency distributions in panels (b) and (d) regardless? This would lend visual symmetry to the plot and provide an alternative perspective on departures from the Gaussian/beta fits.

At the risk of offering more than is needed on the matter, here is an abbreviated answer to the question. First, the key point of these figures is to illustrate the differences between the spheres and the natural particles, and the rounded and angular particles, where the precise distributions involved are of secondary concern. Second, including

empirical frequency distributions (which we take to mean histograms) in panels (b) and (d) would only provide redundant information with panels (a) and (c). The associated probability densities are provided in (b) and (d) for visual reference. Histograms certainly can be valuable for conveying information, and plotting continuous probability density functions together with histograms for the purpose of assessing "fit" can be qualitatively valuable in initial descriptive analyses. But regarding the latter, we prefer to avoid this practice beyond initial assessment unless the number of data $N$ is exceedingly large. Preferred presentations involving continuous random variables include cumulative and exceedance probability plots, quantile-quantile plots, etc., because these do not depend on the choice of bin sizes and are far more sensitive to distribution tails. We also note that the cumulative distribution function (and its fitting) supersedes the probability density function in probability distribution theory because of the nuances that go with the mathematical definitions of probability and "density" in relation to the fact that the real number line (continuum) is not a countable set. Discrete probability mass functions, on the other hand, are a different matter.

209: "Between" these figures?

Corrected

L232-L235: This along with Fig. 5 is a nice explanation of how roughness encourages non-collinear collisions, which then give rise to torques about the center of mass. Yet the continuation of this explanation from L238-249 seems to add complication without producing additional insight.

After reading through again several months later, we agree that this paragraph is not needed.

L255: Videos are great !

Thank you : )

Figure 7: If it's easy enough, you might modify the y-limits on the S=0.28 (bottom right) panel to remove the excessive white space.

We prefer to leave the white space for effect.

Figure 7 caption: The concavity of these semi-log plots indicate whether particles are accelerating or decelerating – or am I wrong? Thermal collapse = deceleration toward zero velocity?

The concavity reflects the value of the shape parameter $A$. The concavity is negative with $A < 0$ (net cooling), it is zero with $A = 0$ (isothermal), and it is positive with $A > 0$ (net heating). Associating net cooling with deceleration and net heating with acceleration is deterministically correct for the motion of a single particle. But this association is incorrect when describing the probabilistic behavior of an ensemble (cohort) of particles. With the onset of particle-surface collisions, the probability density function $f_{E_p}(E_p, x)$ of particle energy states $E_p$ cannot be a Dirac function. That is, the variance of this distribution cannot be zero. A particle at any energy state $E_p$ can become disentrained within a small interval $\mathrm{d}x$. Nonetheless, particles at low energy states are preferentially disentrained relative to particles at large energy states. This

means that, by definition, deposition gives a positive contribution to the average particle energy $E_a$. As fully described in Furbish et al. (2021a), this effect is entirely analogous to the results of Brilliantov et al. (2018) wherein the average energy of a dissipative granular gas increases due to particle aggregation while the total energy of the gas decreases. Indeed, because of this effect of deposition, isothermal conditions ($A = 0$) require an overall (on average) deceleration of the particles. Even with net heating ($A > 0$), particles may on average be decelerating with a positive contribution to the average energy associated with deposition. In contrast, particles that experience net cooling ($A < 0$) must also be decelerating. By definition all particles that are deposited must ultimately decelerate to zero velocity regardless of their energy/velocity histories prior to deposition. The simple description that thermal collapse represents deceleration to zero velocity therefore is an inadequate description of the behavior of the particle ensemble. Disentrainment during net cooling is an inhomogeneous Poisson process involving the ultimate deceleration of all particles to zero velocity. But likewise, net heating involves the ultimate deceleration of particles to zero velocity upon deposition. Both cases lead to specific forms of the distribution of travel distances.

**We have added clarification in the text centered on the first part of the paragraph above. This includes replacing the balance involving the total energy $E$ with the balance involving the average energy $E_a$. The last term in this equation represents apparent heating associated with deposition.**

L291: generalized

Corrected

L294: high-speed, particles launched

Corrected

L295: Pareto

Corrected

Table 3: high-speed: actually you should search for all missing hyphenations. There are more.

Corrected

L336: Not sure how the "dictated by the geometry of the porous medium..." is relevant here. Are you presenting diffusion in porous media as an analogue of top-down diffusion? If so this was not clear to me. I am also confused by your distinction between dispersion and diffusion, or are you using these terms interchangeably? It seems up to now the word "diffusion" has been avoided, but later it gets drawn in with relation to Seizilles et al 2014.

Here we are presenting flow through a porous medium as an analogue to bottom-up diffusion where characteristics of the medium, or the surface in our case, determine diffusion behavior. This is distinct from the top-down influence of particle angularity, which has no analogue in the porous flow example.

We have modified our wording to better accentuate "diffusion" rather than "dispersion" although we are using them interchangeably. We note that "dispersion" often is used to describe macroscopic spreading behavior in order to avoid confusion with molecular diffusion, as in porous media transport problems. But often "diffusion" is meant as a generic mathematical description of spreading behavior regardless of scale. Also see our response below.

Figure 11. This figure appears twice (although I expect this would not survive the copy-edit anyway)

It is a continuation of the same figure (Slopes 0.00 and 0.09 for the first appearance, Slopes 0.15 and 0.18 for the second). The separation into two parts is indeed clunky and has been corrected in the two-column manuscript.

L399: Wouldn't Fickian have a slope of 1?

The formulation of transverse diffusion in Eq. (15) involves the cumulative variance $\sigma_y^2(x)$ rather than the local variance as normally envisioned. We thus recognize that clarification is needed.

Let $s_y^2(x)$ denote the "usual" local variance as this increases with position $x$. In this situation, Fickian diffusion is described by the Einstein-Smoluchowski equation applied to space (rather than time). Namely,

$$\frac{\mathrm{d}s_y^2(x)}{\mathrm{d}x} = 2\kappa_y \,, \tag{1}$$

where $\kappa_y$ [L] is the transverse (spatial) diffusivity. This leads to

$$s_y^2(x) = 2\kappa_y x \,, \tag{2}$$

which, upon taking logarithms, is a straight line with a slope of one in log-log space. However, the cumulative variance $\sigma_y^2(x)$ is

$$\sigma_y^2(x) = \int_0^x s_y^2(x') \, \mathrm{d}x' = \kappa_y x^2 \,. \tag{3}$$

Taking logarithms leads to a straight line with a slope of two in log-log space. The cumulative variance, if associated with Fickian behavior, would have a slope of two. Incidentally, this is a good example of the use of "diffusion" as a mathematical concept applied to the spreading of particles at a decidedly macroscopic scale.

**We have elaborated the formalism to clarify these points in the text.**

Figure 12. S=0.18 panel subtitle you can see little accidental icons from your image-editing software.

Good catch!

L425: Particle properties, ... , influence ... (not influences)

Corrected

L535: Chartrand ?

Now that's embarrassing.